



# New particle formation and its effect on CCN abundance in the summer Arctic: a case study during PS106 cruise

Simonas Kecorius[1], Teresa Vogl[1,4], Pauli Paasonen[2], Janne Lampilahti[2], Daniel Rothenberg[3], Heike Wex[1], Sebastian Zeppenfeld[1], Manuela van Pinxteren[1], Markus Hartmann[1], Silvia Henning[1], Xianda Gong[1], Andre Welti[1], Markku Kulmala[2], Frank Stratmann[1], Hartmut Herrmann[1], and Alfred Wiedensohler[1]

[1]Leibniz Institute for Tropospheric Research (TROPOS), 04318 Leipzig, Germany
[2]Department of Physics, University of Helsinki, P. O. Box 64, 00014 Helsinki, Finland
[3]ClimaCell, Inc., Boston, 02210 Massachusetts, USA
[4]Institute for Meteorology, University of Leipzig, D-04103 Leipzig, Germany

*Correspondence to*: Simonas Kecorius (kecorius@tropos.de)

**Abstract.** In a warming Arctic the increased occurrence of new particle formation (NPF) is believed to originate from the declining ice coverage during summertime. Understanding the physico-chemical properties of newly formed particles, as well as mechanisms that control both particle formation and growth in this pristine environment is important for interpreting aerosol-cloud interactions, to which the Arctic climate can be highly sensitive. In this investigation, we present the analysis of NPF and growth in the high summer Arctic. The measurements have been done on-board Research Vessel Polarstern during the PS106 Arctic expedition. Four distinctive NPF and subsequent particle growth events were observed, during which particle (diameter in a range 10-50 nm) number concentrations increased from background values of approx. 40 up to 4000 cm$^{-3}$. Based on particle formation and growth rates, as well as hygroscopicity of nucleation and the Aitken mode particles, we distinguished two different types of NPF events. First, some NPF events were favored by negative ions, resulting in more-hygroscopic nucleation mode particles and suggesting sulfuric acid as a precursor gas. Second, other NPF events resulted in less-hygroscopic particles, indicating the influence of organic vapors on particle formation and growth. To test the climatic relevance of NPF and its influence on the cloud condensation nuclei (CCN) budget in the Arctic, we applied a zero-dimensional, adiabatic cloud parcel model. At an updraft velocity of 0.1 m s$^{-1}$, the particle number size distribution (PNSD) generated during nucleation processes resulted in an increase of the CCN number concentration by a factor of 2 to 5, compared to the background CCN concentrations. This result was confirmed by the directly measured CCN number concentrations. Although particles did not grow beyond 50 nm in diameter and the activated fraction of 15-50 nm particles was on average below 10%, it could be shown that the sheer number of particles produced by the nucleation process is enough to significantly influence the background CCN number concentration. It implies that NPF can be an important source of CCN in the Arctic. However, more studies should be conducted in the future to understand mechanisms of NPF, sources of precursor gases and condensable vapors, as well as the role of the aged nucleation mode particles on Arctic cloud formation.


## 1 Introduction

Atmospheric new particle formation (NPF), during which particles with diameters from 1 to 2 nm are formed, is a phenomenon observed in many different environments around the world (Kerminen et al., 2018). Initial steps involved in particle formation and subsequent growth are usually clustering and condensation of both organic and inorganic vapors (Schobesberger et al., 2013). Ions are also known to be involved in the nucleation process (e.g. Jokinen et al., 2018). If newly formed particles are not lost due to coagulation (Lehtinen, et al., 2007), and manage to grow to sizes >50 nm, they can

act as cloud condensation nuclei (CCN, Kerminen et al., 2012). Under the presence of sufficient water vapor, CCN activate to form cloud droplets (Köhler, 1936). Atmospheric NPF is estimated to be a substantial source of the world's CCN budget (Merikanto et al., 2009). Thus, in a highly sensitive atmosphere such as the Arctic, where CCN number concentration is usually low (< 100 cm$^{-3}$, Mauritsen et al., 2011), NPF may be an important phenomenon controlling the radiative forcing (Allan et al., 2015).

During the last decade, Arctic regions have experienced remarkable changes. Here, the near-surface temperature has increased at least twofold compared to the Northern Hemisphere (a phenomenon known as Arctic amplification, Overland et al., 2011; Jeffries and Richter-Menge, 2012). In parallel, a substantial decline in multiyear sea ice cover (e.g. Bi et al., 2018), an increase in sea ice mean speed and deformation (Rampal et al., 2009), development of melt ponds (Polashenski et al., 2017), etc., was also observed. Such changes do not only reflect in the dynamics of the Arctic ecosystem (Meier et al., 2014),

but are also predicted to impact mid-latitude climate (Serreze and Barry, 2011; Cohen et al., 2014; Walsh, 2014).

Recent studies suggest that the amplified warming in the Arctic and related changes are a result of a complex interaction between different feedback mechanisms including parameters such as temperature (Pithan and Mauritsen, 2014), surface albedo (e.g. Screen and Simmonds, 2010; Taylor et al., 2013), water vapor (Graversen and Wang, 2009), cloud (Vavrus, 2004), and the lapse-rate (Bintanja et al., 2012). Additionally, variations in atmospheric and oceanic heat transport were also

identified as active players in the changing Arctic climate (Spielhagen et al., 2011; Alexeev and Jackson, 2013). Increase in latent heat and moisture transport towards the poles may drive the low-cloud formation, and thus, Arctic surface warming (Praetorius et al., 2018). And while the mechanisms of lapse rate, surface albedo, temperature and water vapor feedbacks are rather well understood, the net cloud feedback still retains one of the largest uncertainties (Zhang et al., 2018).

Atmospheric studies in the Arctic, although present, are limited due to high costs of logistics and challenging environmental

conditions in the regions (e.g. Willis et al., 2017; Wendisch et al., 2018). This is also valid for NPF studies. Measurements of ultrafine particle physico-chemical properties in the Arctic region were identified as an important aspect to better understand aerosol-cloud-climate interactions (Willis et al., 2017). Although the frequency of atmospheric NPF event occurrences is expected to increase due to Arctic sea ice melt (Dall'Osto, et al., 2017), there is only a limited number of studies that focus on nucleation mode particles in this remote environment. For example, Wiedensohler et al. (1996) reported the occurrence of



ultrafine particles in the Arctic as a result of NPF. However, no correlation with potential precursor gases has been found. Karl et al. (2012) found that a sulfuric acid nucleation mechanism best explains the observed growth of nucleation mode particles over the central Arctic Ocean. In another study by Karl et al. (2013), marine granular nanogels were proposed as a novel route to atmospheric nanoparticles in the high Arctic. Furthermore, NPF in the Arctic region was associated with marine biological processes, such as the seasonal cycle of the gel-forming phytoplankton by Heintzenberg et al. (2017).

Iodine from coastal macro algae was detected in the growing particles (Allan et al., 2015; Sipilä et al., 2016), suggesting the iodine as a nucleation precursor. Croft et al. (2016) showed that ammonia from seabird-colony guano is a key factor contributing to bursts of newly formed particles at Alert, Nunavut, Canada. From the results of volatility measurements, Giamarelou et al. (2016) have proposed that particles during NPF events in the high Arctic exist in the form of partly or fully neutralized ammoniated sulfates. Aerosol particle growth in the Canadian Arctic Archipelago during summer was correlated

with organic species, trimethylamine, and methanesulfonic acid (MSA), suggesting an important marine influence (Willis et al., 2016, Abbatt et al., 2019). This was further supported by Park et al. (2017), who provided a compelling evidence of the contribution of marine biogenic dimethyl sulfide (DMS) to the formation of aerosol particles. On the other hand, studies on whether nucleation mode particles (diameter of 20 nm) can act as CCN are even scarcer. Leaitch et al. (2016) investigated effects of 20–100 nm particles on liquid clouds in the clean summertime Arctic and found that particles as small as 20-50 nm

can activate to cloud droplets. In the pristine environment, where cloud radiative forcing is limited by CCN available (Mauritsen et al., 2011), information about aerosol sources is crucial in understanding the link between sea ice melt and low altitude clouds.

In this investigation, we analyzed four cases of NPF and a subsequent growth from a perspective of particle physical (number concentration, number size distribution, and formation and growth rates) and indirect-chemical (hygroscopicity)

properties. Our main goal here is to test the hypothesis that NPF and secondary aerosol production can influence the CCN budget in the summertime Arctic. The study is structured as follows. After a short description of materials and methods in section 2, we proceed by describing each NPF event separately (section 3). This includes specification of the meteorological conditions during which NPF occurred, characterization of particle formation and growth rates, followed by the observed hygroscopicity of newly formed particles, and the measured CCN concentrations during NPF events. We start the discussion

of the results (section 4) with general overview of our observations, putting the results into perspective of other studies. This leads to section 4.1, where we discuss the indirect evidence of the composition of newly formed particles. Here, we reflect on our observational data as well as various techniques to gain information on particle formation mechanisms, possible sources of precursor gasses, etc. The discussion session is closed by investigating the implication of NPF for cloud formation. This is done by using zero-dimensional parcel model to examine, whether newly formed and slightly grown

particles can become CCN. Model results are compared to measured number concentration of CCN during the NPF events. Main results are summarized at the end of the work, general conclusions are also provided.



## 2 Materials and methods

### 2.1 Description of observations

The data used in this study were obtained during two legs of an expedition of the German Research Vessel Polarstern (PS
106/1 and PS 106/2): the "Physical feedbacks of Arctic boundary layer, Sea ice, Cloud and AerosoL (PASCAL, PS 106/1)"
and " Survival of Polar Cod in a Changing Arctic Ocean (SiPCA, PS 106/2)" (Macke and Flores, 2018; Wendisch et al.,
2018). Both expeditions took place in the vicinity of Svalbard (Norway) from May to July, 2017. PASCAL was performed in
the framework of the ArctiC Amplification: Climate Relevant Atmospheric and SurfaCe Processes, and Feedback
Mechanisms (AC)[3] project and was designed to explore cloud properties, aerosol impact on clouds, atmospheric radiation
and turbulent-dynamical processes. During the first leg of the trip (PS 106/1, PASCAL), the RV Polarstern reached approx.
82 degrees north where an ice-floe camp was established (5 – 14 June). The first leg of the expedition ended at
Longyearbyen, Svalbard by the 21 June. On the 22 June, RV Polarstern left Svalbard for the SiPCA expedition. On second
expedition leg aerosol particle measurements were performed until 16 July. The cruise track and the ice-drift are shown in
Fig. 1.

### 110 2.1.1 Ship deck observations during NPF events

#### 2.1.1.1 31 May – 1 June (NPF 1)

RV Polarstern arrived at the marginal ice zone on 31 May 2017 and entered the pack ice around 3 pm (note that all times in
this study are given in UTC). In this area, the ice was broken up by leads, which facilitated the passage of the vessel towards
the north. Around 8 pm a region with more densely packed ice was reached, which obstructed the movement of the ship
115 (Nicolaus, 2018). On these occasions, due to frequent reverse-forward ship movement, pollution highly affected the
measurements on-board. On 1 June, the vessel could once again pass through open leads in the pack ice, allowing for
contamination-free scans for the time period from 4 am to 8 pm. During this time, RV Polarstern moved 26 km (from
80.39°N 7.58°E to 80.62°N 7.94°E) in mostly cloud-free conditions. From 6 pm to 8 pm, a thin ice cloud was present in over
8 km altitude. Also, over a short period from 2 to 3 pm, intermittent low-level liquid clouds were present, which however did
not decrease the global radiation significantly.

#### 2.1.1.2 17 – 19 June (NPF 2)

On 17 June, the ship was moving southward through packed ice area, breaking floes and navigating through polynyas
(Nicolaus, 2018). Over the complete day of 17 June, low-level stratocumulus clouds were present, which were broken up
occasionally between 7 am and 1 pm, and 4 to 10 pm. Between 11 pm on 17 June and 1 am on 18 June, measured visibility
decreased, accompanied by an increase in relative humidity (RH), indicating fog. This low-level cloud layer was present





until approximately 8 am on 18 June. At around the same time, Polarstern reached the ice edge. During the following 9 hours, until 6 pm, no clouds were present except for a very thin, high ice cloud at 8 km from approximately 11:30 am to 12 pm. This period of high incident radiation was only briefly interrupted by a short fog event from 3 to 3:30 pm. During this whole time, RV Polarstern moved through open water, but was always surrounded by floating ice. Starting at 6 pm, a thin

low-level cloud layer was present above the ship, which decreased the global radiation significantly. This cloud layer was present until the next day, 19 June, at approximately 12 pm. During 19 June, RV Polarstern was moving through open water and ice along the west coast of Spitsbergen Island (Fig. 1). From approximately 12:30 pm to 3 pm another short cloud-free period led to high global radiation. At 4 pm at approx. 3 km altitude a cloud moved in decreasing the global radiation once again.

**2.1.1.3 25 – 28 June (NPF 3)**

The third NPF and growth event analyzed in this work occurred during the second leg of the expedition, when RV Polarstern was East of Svalbard, moving towards the North. During the complete period of interest, the ship was very close to the ice edge (Fig. 1). Areas dominated by open water were passed by the vessel, as well as ice-covered water (Nicolaus, 2018). However, the ice was never very densely packed and the transit of the ship did not require breaking the ice. Low-level clouds

and fog were present during all of 25 June to 27 June; on 28 of June a short period of cloud-free conditions was observed from around 4 to 6 am. There were two short floe stations, one on 25 June from around 5 pm until midnight and the other on 27 June from around midnight to 3 am.

**2.1.1.4 1 – 3 July (NPF 4)**

From midnight of 1 July to 4 July 0 am, RV Polarstern was moving northwards from 81.64°N 32.62°E to 82.16°N 32.87°E.

This region was mostly ice-covered with some open leads through which the vessel could pass without having to break the ice. At this time of the expedition, melt ponds were observed frequently on the ice floes. On 1 July, there was a thick (up to 3 km altitude) low-level cloud layer present until 2 pm associated with some snow fall. After 1 pm, the cloud bottom height increased steadily; however, some intermittent fog was still present at sea level. A single fogbow was observed between 6:20 and 7 pm. The fog dissolved at midnight on 2 July. Almost throughout the entire day of 2 July, no clouds were present

except for optically thin cirrus clouds, allowing for high solar irradiation. For a more detailed description of local and associated large scale weather patterns during PS106 refer to Knudsen et al. (2018).

**2.2 Measurement setup and equipment**

To measure aerosol particle physico-chemical properties, a temperature controlled measurement container, prepared and operated by the Leibniz Institute for Tropospheric Research, Leipzig, Germany, was installed on the observation deck of RV

Polarstern. The aerosol container was air-conditioned to 24 °C and the aerosol inlet head was heated to 30 °C to ensure the





stability of aerosol instrumentation and prevent icing, respectively. The aerosol inlet was made of 6 m length stainless steel tubing, with an inner tube diameter of 40 mm. It was placed on top of the measurement container with an angle of 45 degrees, pointing away from the ship. The aerosol flow in the 6 m long inlet was set to 40 l/min (Reynolds number <2000, laminar flow) to minimize particle losses. Inside the container, an isokinetic splitter was used together with short and vertical

conductive tubes to feed the measurement instrumentation with an aerosol sample. Aerosol instrumentation (relevant to this study) included a neutral cluster and air ion spectrometer (NAIS), a mobility particle size spectrometer (MPSS), Volatility/Hygroscopicity-Tandem Differential Mobility Analyzer (VH-TDMA), and Cloud Condensation Nucleus Counter (CCNC) to measure aerosol particle number size distribution, volatility/hygroscopicity properties of aerosol particles, and the number concentration of CCN, respectively.

**2.2.1 Neutral cluster and air ion spectrometer (NAIS)**

Neutral cluster and air ion spectrometer (NAIS, Mirme and Mirme 2013) and guidelines by Kulmala et al. (2012) were used to study early stages of NPF and subsequent growth (including NPF event classification, formation ($J$) and growth rate (GR) calculation). NAIS measures number size distribution of neutral particles in the diameter range of approx. 2 – 40 nm and charged particles and clusters in the size-range of approx. 0.8 – 40 nm. The instrument is an extended version of the air ion

spectrometer (Mirme et al., 2007) and utilizes a sample preconditioning section to enable measurements of neutrally charged particles. Unipolar corona chargers are used for both charging and charge neutralization. Charged particle classification is carried out in the multichannel differential mobility analyzer (DMA) where 21 individual electrometers are used to record electric current carried by the charged particles. Due to high total flow of NAIS (60 L min$^{-1}$) a dedicated 1.3 m long copper inlet (3.5 cm in diameter) was installed to sample ambient air. Measurement data were inverted using the v14-lrnd inversion

algorithm (Wagner et al., 2016). Particle losses due to diffusion were corrected before data processing.

**2.2.2 Mobility particle size spectrometer (MPSS)**

Particle number size distributions (PNSD), in a mobility size range from 10 to 800 nm, were measured with a TROPOS-type mobility particle size spectrometer (MPSS, Wiedensohler et al., 2012). The MPSS consisted of a Hauke-type DMA (effective length of 28 cm), condensation particle counter (CPC, model 3772, TSI Inc., USA, flow rate 1 L min$^{-1}$), a closed-

loop sheath flow arrangement and a bipolar diffusion charger, assuring the bipolar charge equilibrium as described in Wiedensohler (1988). The sample flow rate was controlled by a CPC (1 L min$^{-1}$) and the sheath flow rate was 5 L min$^{-1}$. The time resolution of an up-and-down scan was 5 min. Electrical particle mobility distributions were inverted to PNSDs using the inversion algorithm presented by Pfeifer et al. (2014). The final PNSDs were corrected for transmission losses in the sampling lines using the method of equivalent length and CPC counting efficiencies (Wiedensohler et al., 1997). Sizing

accuracy of MPSS was controlled using nebulized polystyrene latex spheres (PSL, Thermo Scientific™, Duke Standards™)





of 203 nm (Wiedensohler et al., 2018). High voltage supply offset calibration, instrument flows and tests for leakage were performed on a regular basis (once per week).

### 2.2.3 Volatility/Hygroscopicity tandem differential mobility analyzer (VHTDMA)

Aerosol particle affinity to water and volatility properties (not discussed here) were measured using the TROPOS-type

Volatility/Hygroscopicity tandem differential mobility analyzer (VHTDMA, Augustin-Bauditz et al., 2016). The instrument consists of a DMA-1 that selects chosen quasi-monodisperse particles, a thermodenuder (not used in this study), an aerosol humidification section that conditions the particles selected by the DMA-1, and a MPSS-equivalent closed-loop sheath flow unit inside the temperature controlled box, which is used to obtain the hygroscopic growth factor (HGF). The HGF is defined as the ratio between the measured particle electrical mobility diameter at a given RH as measured by the second DMA and

the initially selected, dry diameter.

During the whole expedition, two constant aerosol particle sizes, 50 and 150 nm, were selected for the measurement of HGF at a target RH of 90%. Additionally, HGF of 15, 20 and 30 nm size particles were measured during NPF and growth events. The system RH, measured by a humidity sensor, was periodically calibrated by an automatic calibration unit, using pure ammonium-sulfate. Scans with RH ±2% from target RH were excluded from data analysis. Sizing accuracy, high voltage

supply offset calibration, flow rates, and zero tests were performed regularly (once per week). In general, recommendations have been followed as described in Massling et al. (2011).

The VH-TDMA data was inverted using a TDMAinv routine (Gysel et al., 2009) to retrieve the Probability Density Functions of GF (GF-PDF). Scans with RH < 20% were used to calibrate size offset in the system, as well as to define the width of the transfer function (Gysel et al., 2009). Particle hygroscopicity was derived based on $\kappa$-Köhler theory following

Petters and Kreidenweis (2007).

### 2.2.4 Cloud condensation particle counter (CCNC)

The CCNC (model CCN-100 from Droplet Measurement Technologies, Roberts & Nenes, 2005) measured CCN number concentrations, subsequently at six different supersaturations (0.1, 0.15, 0.2, 0.3, 0.5 and 1%), where each supersaturation was sampled for 10 minutes. Hence an hourly average concentration at each supersaturation is available. The instrument was

calibrated before and directly following the campaign using pure ammonium sulfate particles of known sizes, based on the ACTRIS protocol (Gysel & Stratmann, 2013).



### 2.2.5 Offline chemical analysis

The sampling of aerosol particles was conducted using five-stage low-pressure Berner impactors (Hauke, Austria) with 50% cut-offs at 0.05, 0.14, 0.42, 1.2, 3.5, and 10 µm aerodynamic diameter and a flow rate of 75 L min$^{-1}$, which was installed on

the top of the observation deck facing the ocean at a height of ca. 25 m. The aerosol particles were collected on aluminum foils as impaction substrates, which had been heated at 350 °C for at least 2 hours to reduce blank levels prior to sampling. To avoid condensation of atmospheric water on the surface of these aluminum foils, a conditioning unit was mounted between the impactor inlet and the sampling unit consisting of a 3 m tube. By heating the sampled air, high relative humidity of the ambient air was reduced to 75-80% before the collection of the aerosol particles. The temperature difference between

the ambient air at the impactor inlet and the sampled air after the conditioning unit did not exceed 9 K. After sampling, the aluminum foils were stored in aluminum boxes at -20 °C and transported in dry ice to the TROPOS laboratories in Leipzig, Germany. Field blanks were collected by loading the Berner impactor with the aluminum foils at the sampling site with no air drawn through it. Please note that the sampling time was set to 72 or 144 hours (to accumulate enough particle mass on the filters), thus, it does not exclusively comprise the discussed NPF events. For example, during NPF Event 1, chemical

particle composition was determined from samples that were collected between 29 May (midday) and 1 June (approx. 8 am.). During the NPF Event 3, sampling was done between 25 June (11 am) to 28 June (9 am).

Particle mass determination was performed by weighing clean (blank) and particle-loaded filters using a microbalance UMT-2 (Mettler-Toledo, Switzerland). The concentrations of water-soluble methanesulfonic acid (MSA) and inorganic compounds relevant to this study ($SO_4^{2-}$, $NH_4^+$, $Na^+$) in filtered (0.45 µm syringe) aqueous extracts (50% of the filter in 2 mL) were

determined using ion chromatography (ICS3000, Dionex, Sunnyvale, CA,USA), as described in Müller et al. (2010). Assuming that the ocean is the major source of the measured atmospheric sodium, sea salt sulfate (ss-sulfate) was calculated from the constant mass ratio ($\frac{SO_4^{2-}}{Na^+} = 0.251$) in bulk seawater (Galloway et al., 1993; Fomba et al., 2011). Non-sea salt sulfate (nss-sulfate) was calculated by subtracting ss-sulfate from the total sulfate concentration. The determination of total carbon (TC) as organic carbon (OC) and elemental carbon (EC) was carried out by a two-step thermographic method (C-mat

5500, Ströhlein, Germany) with nondispersive infrared sensor (NDIR) detection as described in Müller et al. (2010). Organic matter (OM) was calculated by considering OM as twice OC (OM=2 x OC) for remote aerosols (Turpin and Lim, 2001)

### 2.3 Analysis of PNSD measurements

Before NPF event classification, inverted and loss-corrected NAIS and MPSS PNSDs were merged together. For the smallest particle diameter, from 2 to 10 nm, exclusively NAIS data was chosen. This is because the MPSS used in this study

was optimized to operate in a diameter range from 10 to 800 nm. The diffusional losses of sub-10 nm particles were too great to accurately recover the PNSD at initial steps of nucleation. Contrarily, uncertainties in the NAIS measured particle number concentration increases for particle diameters larger than 10 nm (Wagner et al., 2016). For these reasons, PNSDs





from both NAIS and MPSS were merged at 10 nm diameter. No additional treatment (e.g. spline fit to smooth merging distributions) was performed on merged PNSDs.

Following the protocol by Kulmala et al. (2012), NPF events were visually identified from the merged PNSDs. Although different types of NPF were recorded (e.g. short bursts in the smallest particle number as e.g. described for the Arctic region by Heintzenberg et al., 2017 and Dall'Osto et al., 2017), in this work we will only focus on NPF events with subsequent particle growth. This type of event does not only include particle formation, but also includes later particle growth lasting for several hours, thus representing a more regional phenomenon (Ström et al., 2009). It also allows us to calculate GR of the

particles, which would not be possible in the case of short nucleation mode particle bursts.

Different methods exist to determine the GR based on the measured PNSD. For example, maximum-concentration and log-normal distribution function methods were proposed by Kulmala et al. (2012). Tracking regions of PNSD and interpreting the change rate of the size-integrated general dynamic equation methods was suggested by Pichelstorfer et al (2018). In this work, we used try-error approach to find the best fit to determine GR by selectively applying all mentioned methods for

certain NPF cases. The formation rate of particles of certain size ($J$) was calculated as described by Kulmala et al. (2012), based on the observed changes in particle concentrations, determined GR, and particle losses characterized by coagulation sink (CoagS).

## 2.4 Adiabatic cloud parcel model

To study the climatic relevance of NPF in the Arctic, we have used a zero-dimensional, adiabatic cloud parcel model.

Thorough formulation of the model is given by Rothenberg and Wang (2016) and will not be discussed here. Model code is also freely available at https://pyrcel.readthedocs.io. Shortly, at the initial step, the model calculates an equilibrium wet-size distribution from the set of given parameters. This includes the description of the aerosol population and environmental specifications of temperature, pressure, relative humidity, parcel ascending velocity, and the height of the planetary boundary layer. The aerosol particle population, consisting of two modes, is described by the total number concentration, the

geometric mean diameter, and the geometric standard deviation of the log-normal distribution. The hygroscopicity parameter $\kappa$ following Petters and Kreidenweis (2007) is used to describe particle chemical composition. The evolution of the parcel supersaturation, temperature, pressure and, liquid/vapor water content are then integrated forward in time to describe the thermodynamic evolution of an adiabatically lifted, non-entraining parcel. In the model, the evolution of supersaturation $S$ is:

$$\frac{dS}{dt} = \alpha(T,P) - \gamma(T,P)\frac{dw_c}{dt}, \tag{1}$$

where $\alpha$ and $\gamma$ are functions depending on temperature and pressure (Leaitch et al., 1986) and $w_c$ is the liquid cloud water mass mixing ratio. Change in temperature is described as:





$$\frac{dT}{dt} = -\frac{gV}{c_p} - \frac{L}{c_p}\frac{dw_v}{dt}.$$
(2)

$V$ is the updraft velocity, $g$ – gravity, $c_p$ - is the specific heat of dry air at constant pressure, $L$ is the latent heat of water, and $w_v$ – water vapor mass mixing ratio. Water mass conservation is ensured as vapor condenses into cloud water. Pressure

change within the ascending parcel can be written as:

$$\frac{dP}{dt} = -\frac{gVP}{R_d T_v},$$
(3)

where $T_v$ is temperature, $R_d$- gas constant for dry air. The change in cloud water:

$$\frac{dw_c}{dt} = \frac{4\pi\rho_w}{\rho_a}\sum_{i=1}^{n} N_i r_i^2 \frac{G}{r_i}\left(S - S_{eq}\right).$$
(4)

Here $\rho_a$ and $\rho_w$ is the density of air and water, respectively. $N_i$ is a number concentration and $r_i$ is radius in a size bin, $S$ is

environmental saturation, $S_{eq}$ is the predicted equilibrium supersaturation under framework described by Petters and Kreidenweis (2007). $G$ is a growth coefficient, which is a function of both the chemical and physical properties of particles.

**3 Results**

During the PS106 cruise, a number of instances were recorded whereby a total particle number concentration (integrated from MPSS between 10 to 800 nm) suddenly increased from the background concentrations of several hundred to several

thousand particles per cm3 (Fig. 1). After eliminating the contribution from the ship exhaust (by filtering abrupt and short increases in particle number concentration recorded by a total CPC with 1 second time resolution), most of the cases when the particle number concentration increased tenfold can be associated with new particle formation (NPF) events. For further discussion, we have selected four NPF events with a subsequent particle growth, which represent the phenomenon on a regional scale (Ström et al., 2009). To gain information about the scale of NPF, also additional data of PNSD information

from the Villum Research Station and Zeppelin mountain Observatory were taken into account (data for visual inspection were taken fromhttp://ebas.nilu.no/).

The geographic location of the observed NPF events can be seen in Fig. 1 (indicated with black rectangles and date of occurrence), and took place between 78.55 to 81.66 degrees North and 7.28 to 33.96 degrees East. The most intense event (NPF 1) occurred on 1 June 2017, with the total particle number concentration increasing from 100 to more than 4000

particles cm$^{-3}$. During the NPF event, the lateral distance between RV Polarstern and the nearest coast of Svalbard archipelago was 150 km. The least intensive NPF event (NPF 3) was recorded on 26 June during which the total particle number concentration increased from 160 to 700 particles cm$^{-3}$. Nevertheless, the subsequent particle growth from 3 nm to approx. 50 nm lasted for 3 days. All the events that were recorded during June (1st, 18th and 26th) took place in the vicinity





of the marginal ice zone. The most northern event (NPF 4, 81.6 degrees North) was observed on 2 July, 2017. At this time,
the RV Polarstern was further away from marginal ice zone. The average total particle number concentration before the NPF
event was approx. 100 particles cm$^{-3}$, which increased to 1400 particles cm$^{-3}$ during the event.

### 3.1 Overview of the NPF events

In this paragraph, a detailed overview of the events is presented with the focus on environmental conditions during which
NPF occurred, as well as the formation and growth rates of newly formed particles.

### 3.1.1 NPF 1: 1 June

The first NPF event with a subsequent particle growth was observed from around 6 am onwards on 1 June, 2017. The RV
Polarstern reached the marginal ice zone at 11 am on 31 May, 2017. This can be seen from the air and water temperature
profiles (Fig. 2). The temperature of air and water decreased from approx. +5 °C to -5 °C (air) and -2 °C (water). Before the
NPF event, the average particle number concentration in a size range from 10 to 50 nm (PNC$_{10-50}$), was 50 particles cm$^{-3}$.
The particle number concentration in the size range from 100 to 800 nm (PNC$_{100-800}$) before the event decreased from 150 to
as low as 2 particles cm$^{-3}$. This resulted in a sharp decrease in the coagulation sink for 3 nm particles from $7.6 \times 10^{-5}$ s$^{-1}$ to
$8.6 \times 10^{-6}$ s$^{-1}$. Condensation sink also decreased by one order of magnitude from $2.2 \times 10^{-2}$ to $2.2 \times 10^{-3}$ s$^{-1}$, creating
favorable conditions for particles to form. The NPF event occurred when the RH was approx. 90% and particle formation
rate peaked when the global radiation was approaching the maximum (600 W m$^{-2}$). The wind speed gradually decreased
from average 8 on 31 May, 2017, to 5 m s$^{-1}$ during the NPF event. As a result of the NPF, the number of ultrafine particles
increased by almost 2 orders of magnitude.

The backward air mass trajectories (calculated for 200 and 2000 m above sea level, Draxler and Rolph, 2012) showed
possible intrusion of air from higher altitudes, also that air was arriving at the ship following the 80 degrees North latitude,
passing over the Prince George Land and North-East Svalbard archipelago (Fig. 1). This can be confirmed by the increase in
ozone concentration at Zeppelin observatory (Aas et al., 2018; data available from http://ebas.nilu.no/). Following the NPF
event on 1 June, the wind direction gradually changed from NE to SW, and brought in a sudden fog (at 7 pm , evident from a
steep increase of ambient RH to 100% and a simultaneous decrease in visibility measured by the vessel's meteorology
station). This can be seen as a sharp increase in both air temperature and RH (to over 100%) causing disruption in the PNSD
(onwards from approx. 8 pm., 1 June). At the same time, further observations of the event were corrupted by the local
pollution from ship exhaust.

Some parameters describing newly formed particles and ions are shown in Table 1. The particle GR in a size range from 3 to
7 nm was 1.2 nm h$^{-1}$. After the NPF event, subsequent particle growth lasted for about 12 hours, during which the particles
were able to grow to approx. 30 nm in diameter (geometric mean diameter). The GR for 1.6 to 3 nm ions was somewhat





more variable – 0.7 for negative and 1.4 nm h[-1] for positive ions. Please note that we were not able to calculate the positive

ion GR in a size range from 1.6 to 3 nm. Instead, the GR for a particle size range 1.6 to 4 nm was calculated. The formation

rate of 3-nm ($J_{3-}$) sized neutral particles and negative ions (1.6-nm, $J_{1.6-}$) was approx. 0.4 and 0.045 cm[-3] s[-1], respectively.

### 3.1.2 NPF 2: 18 June

On 18 June, at approx. 6 am. RV Polarstern left the packed ice entering the marginal ice zone, which resulted in water and

air temperature increase from -1.9 °C to approx. 2 and 0.5 °C above zero, respectively. At the same time, local wind speed

decreased from 5 to 2 m s[-1]. The $PNC_{10-50}$ and $PNC_{100-800}$ from 17 June prior to the NPF event were rather stable, with an

average value of approx. 30 cm[-3]. The corresponding coagulation (for 3 nm particles) and condensation sink was $1.2 \times 10^{-5}$

and $2.8 \times 10^{-3}$ s[-1], respectively. Analysis of backward trajectories showed that since midnight of 17 June, air masses were

passing over the Arctic Ocean, and Greenland Sea. From the beginning of 18 June and onwards, air masses were already

passing over the North-East coast area of Greenland (Fig. 1). The NPF event occurred when the global radiation reached its

maximum at 570 W m[-2] and the RH decreased to 85%. During the event, the $PNC_{10-50}$ increased to 3200 cm[-3]. Particle

growth was slightly disturbed by a fog episode (can be seen in PNSD and as RH increase to 100% in Fig. 2) at around 3 pm

and drizzle at 11 pm. Nevertheless, the particle growth remained observable until the evening of 19 June. During this time

(after a period of 32 hours), newly formed particles grew to approx. 50 nm (geometric mean diameter).

The GRs for particles in the size range from 3 to 7 nm were in a range from 3.6 to 4.9 nm h[-1]. The GR for 1.6 to 3 nm

negative ions was 2.9 nm h[-1], and 4 to 9 nm positive ions 3.3 nm h[-1]. The $J_{3-}$ of particles was approx. 0.35 cm[-3] s[-1]. Formation

rate for positive ($J_{1.6-}$) and negative ($J_{4-}$) ions were 0.05 and 0.06 cm[-3] s[-1], respectively. If compared to Event 1, it can be seen

that despite similar intensity of NPF, particle growth during the second event was approx. 2 times faster, and particles were

able to grow to larger diameters (30 nm during event 1 versus 50 nm during Event 2).

### 3.1.3 NPF 3: 28 June

The least intensive NPF event was observed on 26 June. The RV Polarstern was at the marginal ice zone, around 200 km

east of Svalbard. The formation and growth of particles was already observed on both 24 and 25 June during less

pronounced NPF events (not shown), when the ship was approx. 100 km South of Svalbard coast. New particle formation

along the East coast of Svalbard can be seen in Fig. 1 as increase in total particle number concentrations, which were

measured from 24 to 28 June, along a distance of more than 600 km. The daily average of $PNC_{10-50}$ and $PNC_{100-800}$ from June

24 up to the NPF event (26 June) were approx. 600 and 50 cm[-3]. As a result of NPFs on 24 and 25 June, an interesting

pattern emerged in 26 June PNSD (Fig. 2). At the beginning of 26 June (midnight to 3 am), three distinctive modes with

geometric mean diameters of 15, 40 and 150 nm can be seen. The smallest mode at 15 nm is a result of the NPF, which

occurred on 25 June. These newly formed particles slowly grew in size and by 8 am 26 June, the mode at 40 nm emerged,



which was in turn a result of NPF and subsequent particle growth observed on 24 June. Larger size particles (150 nm in

diameter) seem to exist independently from the NPF events, and were present before, during, and after the NPF on 26 June. However, because we were not able to identify particle growth, the NPF events on 24 and 25 June was excluded from the result and discussion sections.

The event on 26 June started with relatively calm winds (2 m s$^{-1}$), which gradually increased to 10 m s$^{-1}$ over a 3 day period (26 to 28 June) with a constant rate of 0.3 m s$^{-1}$ h$^{-1}$. The direction of wind remained stable during the event, with prevailing

winds from South-South West (190° to 200°) direction, and stagnant air masses coming from the marginal ice zone. At the beginning of the event, relative humidity was at around 87%, and remained below 95% during the whole 3 day period. Air and water temperature during the event were approx. -1.5 °C. During the described three day period, water temperature remained the same (with some short episodes of warmer water), while air temperature steadily increased to 0 °C. The NPF event occurred with a global radiation being at its maximum (200 W m$^{-2}$), however, this time solar radiation was at least two

fold lower than observed during previous cases. This is due to the presence of a low-level cloud layer topped at 2 km during the whole day of 26 June. The corresponding coagulation and condensation sink just before the event was $2.2 \times 10^{-5}$ and $6.0 \times 10^{-3}$ s$^{-1}$, respectively.

The GR of 3 to 7 nm particles was in a range from 0.5 to 0.7 nm h$^{-1}$. The GR of negative (1.6 to 3 nm) and positive ions (2 to 6 nm) were accordingly 1.2 and 2.2 nm h$^{-1}$. Despite the noticeable pollution from ship exhaust, particle growth after the NPF

event was observed over the period of three days (Fig. 2). During this time period, particles grew from several nanometers up to sizes of 50 nm (geometric mean diameter). The formation rate of positive ($J_{2-}$) and negative ($J_{1.6-}$) ions were 0.03 cm$^{-3}$ s$^{-1}$, and the $J_{3-}$ for particles was approx. 0.08 cm$^{-3}$ s$^{-1}$.

### 3.1.4 NPF 4: 2 July

On 2 July, the RV Polarstern ventured further into the Arctic ice, more than 300 km from the coasts of Svalbard and Prince

George Land (81.51°N, 32.97°E). The prevailing Western winds were rather stable during a 3 day period (from 1 to 4 June) at 6 m s$^{-1}$. The same was true for water temperature, which remained approx. 2 °C below zero during the whole event period. The air temperature, on the other hand, was varying between negative 1 and negative 5 °C. The calculated backward air mass trajectories indicated that before the midday of 1 July, air was coming from the direction of the Prince George Land. The average PNC$_{10-50}$ and PNC$_{100-800}$ during this time was 60 and 70 cm$^{-3}$, respectively (Fig. 2). From 1 July onwards, air masses

arriving at RV Polarstern passed closer and closer to the North-East coast of Greenland, however, did not pass over the land, as it was the case for Event 2 (Fig. 1). Effective wet removal of particles by fog could be observed during the afternoon hours of 1 July, leading to extremely low particle number concentrations prior to the NPF event. The PNC$_{10-50}$ and PNC$_{100-800}$ respectively decreased to 40 and 10 cm$^{-3}$. The resulting coagulation and condensation sink became $4.5 \times 10^{-6}$ and $1.0 \times 10^{-3}$ s$^{-1}$, respectively. The NPF event started at 8 am on 2 July at an ambient RH of approx. 90%, and a maximum global



radiation of 500 W m$^{-2}$. In parallel to RV Polarstern measurements, the formation of new particles was also observed at both Villum Research Station and Zeppelin Observatory, indicating a regional phenomenon.

The particle GR, in a size range from 3 to 7 nm, was 0.9 nm h$^{-1}$. After 40 hours of growth, the geometric mean diameter of particles reached 30 nm. The GR of negative ions was 1.5 nm h$^{-1}$ (in a size range from 2 to 3 nm). Once again, it has to be noted that for ions, the GR in the 1.6 to 3 nm size range was difficult to obtain. Particle formation rate, $J_{3-}$ was approx. 0.15

cm$^{-3}$ s$^{-1}$. The formation rate of negative ions ($J_{2-}$) was 0.02 cm$^{-3}$ s$^{-1}$. As in the case of Event 1, negative ions seemed to be more prominent than positive ones.

### 3.2 Particle hygroscopicity during NPF events

The size segregated HGF and hygroscopicity parameter $\kappa$ during NPF events is presented in Table 2. Diameters and scan times of dry particles that were selected for HGF measurements are also indicated in Fig. 2. The HGF scans were performed

following the growth of freshly formed particles from MPSS PNSD measurements. In most of the instances newly formed particles grew rather slowly and took between 2 and 7 hours to grow to diameters of 20 - 30 nm, when its HGF was measured. The HGF of 30 - 50 nm particles was measured between 20 to 40 hours after the initial NPF event. Despite the size of selected dry particles, the measured HGF distributions were exclusively mono-modal, indicating internal mixture of the aerosol particle. The highest HGF of nucleation mode particles (15 - 20 nm) was observed during Event 1 and Event 4.

The HGF of 20 nm particles during Event 1 was measured 7 hours after the beginning of the NPF, and was 1.46±0.02 (± standard deviation (sd); $\kappa = 0.41\pm0.02$). At the time of Event 4, HGF of the 15 nm particles was 1.34±0.01 ($\kappa = 0.33\pm0.02$). The lowest HGF of 20 nm particles was observed throughout both Event 2 and Event 3, and was 1.17 ($\kappa = 0.13\pm0.00$) and 1.16 ($\kappa = 0.12\pm0.02$), respectively. Hygroscopicity of slightly grown Aitken mode particle (30 to 50 nm) varied from 1.17±0.02 ($\kappa = 0.11\pm0.00$) to 1.55±0.01 ($\kappa = 0.38\pm0.00$). In general, the longer the particles aged, the more hygroscopic they

became. For example, 8 hours after the new particles were formed during Event 2, the HGF of 20 nm particles was 1.17±0.02. After another 15 hours, these particles grew to sizes of approx. 30 nm, which HGF increased to 1.43±0.05 ($\kappa = 0.36\pm0.08$). Interestingly, the HGF of 50 nm particles was somewhat lower, 1.25±0.01 ($\kappa = 0.16\pm0.04$). Nevertheless, it followed the same pattern and with time increased to the values recorded before the NPF event.

### 3.3 Measured CCN concentrations during NPF events

Concentrations of CCN ($N_{CCN}$) measured during the four NPF events can be seen in Fig. 3. An increase in $N_{CCN}$ during these events can be seen across all supersaturations. To determine the increase, measured data were fitted, visible as lines in Fig. 3. Data included in the fitting were taken from times on when formation rates of particles were noticeably increased (10% of the maximum signal) and go up to the time when the NPF event was interrupted by a change in air mass or fog formation. These periods span 10, 39.5, 44.5 and 29 hours for the NPF events 1 to 4, respectively. Independent of the duration of the





event, the observed increase in $N_{CCN}$ during these periods were mostly roughly a factor of two for supersaturations from 0.1
to 0.5% and roughly a factor of 3 to 6 at 1%. This larger increase at the highest supersaturation is related to the fact that the
number concentrations of smaller particles, which are only activated at higher supersaturations, increased the strongest.
During NPF event 2, the increase was somewhat lower, mostly below a factor of 2. These measurements clearly show that
during NPF events not only new particles are generated, but also that particulate mass is gained on particles of all sizes,

increasing their size and hence their ability to act as CCN at a given supersaturation. A similar observation was made in
Antarctica (Herenz et al., 2019), where NPF events with increases in total particle number concentrations from a few
hundred to thousands of particles per cm$^3$ were also accompanied by an increase in $N_{CCN}$ of at least a factor of 2 at all
examined supersaturations. This is in agreement with modelling results by Merikanto et al. (2009), where CCN in Arctic
regions were found to almost exclusively originate from NPF.

**3.4 Chemical composition of size resolved particles**

The size-resolved absolute atmospheric concentrations of ammonium, MSA and nss-sulfate for the selected periods versus
campaign average are shown in Fig. 4. On average, the highest concentrations of nss-sulfate (81 and 70 ng m$^{-3}$), MSA (18
and 10 ng m$^{-3}$), and ammonium (16 and 8.7 ng m$^{-3}$) were found on the impactor stages 2 and 3, respectively. While the
concentrations of nss-sulfate and ammonium on the impactor samples from 25 to 28 June were comparable to the average

values, the impactor samples from 29 May to 1 June stood out with much higher values, especially in the accumulation mode
(nss-sulfate: 251 and 295 ng m$^{-3}$ and ammonium: 34 and 17 ng m$^{-3}$ on impactor stages 2 and 3, respectively). Also for
smaller particles (stage 1), nss-sulfate was found at a much higher concentration (35 ng m$^{-3}$) than the average (8.3 ng m$^{-3}$).
The highest organic matter (OM) mass concentration were found on stage 2 (106 ng m$^{-3}$) and lowest - on stage 5 (39 ng m$^{-3}$).
OM mass concentration for the period from 25 to 28 June strongly exceeded the average concentration, especially in the

accumulation mode (218 ng m$^{-3}$ and 147 ng m$^{-3}$ for stages 2 and 3, respectively). For a time period from 29 May to 1 June
the OM mass concentration ranged close to the average values.

**4 Discussion**

**4.1 General overview**

Although NPF events in the high Arctic were reported by several studies, there are no observations, which use the same or

equivalent measurement equipment as in this study, able to observe the dynamic changes of the smallest particles (formation
and growth of >1.6 nm clusters). Because of this, we have also calculated the rate at which new particles appear at larger
diameter (10 nm, $J_{10}$). The values of so called apparent nucleation rates are more frequently reported in the literature. For
example, in a several studies from the Svalbard region, GRs for 5 to 25 nm particles were reported to be from 0.1 to 0.6 nm
h$^{-1}$, but in general ≤1.0 nm h$^{-1}$ (Ström et al., 2009; Giamarelou et al., 2016; Heintzenberg et al., 2017). The corresponding



$J_{10-}$ values were in a range from 0.1 to 1.4 cm$^{-3}$ s$^{-1}$. In case of this study, GR for 5 - 25 nm and $J_{10-}$ values varied correspondingly from 0.7 - 5.4 nm h$^{-1}$ and 0.04 - 0.4 cm$^{-3}$ s$^{-1}$, respectively. The GR of 5 - 25 nm size particles in this study was on average 0.9 nm h$^{-1}$. The GR of 5 - 25 nm particles on 18 June, however, outstands other NPF events with GR being significantly higher, 5.4 h$^{-1}$. During the same event, the $J_{10-}$ was also higher, 0.4 cm$^{-3}$ s$^{-1}$. Nevertheless, on average, the observed GR and $J_{10-}$ values were in the same order as reported in other studies from Arctic region (e.g. Asmi et al., 2016).

Some studies for similar environmental conditions also exist. Jokinen et al. (2018) provided a comprehensive study on the particle formation in coastal Antarctica. The growth and formation rates for 3 nm particles were found to be between 0.3 - 1.3 nm h$^{-1}$, and 0.03 - 0.14 cm$^{-3}$ s$^{-1}$. It was concluded that ion-induced nucleation of sulfuric acid and ammonia is a major source of secondary aerosol particles in the pristine Antarctic environment. Kyrö et al. (2013) reported formation rates of negative clusters ($J_{1.6-}$, 0.01 to 0.4 cm$^{-3}$ s$^{-1}$) measured at the Finnish Antarctic Research Station, Aboa, in Dronning Maud

Land. In addition, apparent nucleation rates of 10 nm particles at Aboa ranged from 0.003 to 0.3 cm$^{-3}$ s$^{-1}$. In yet another Antarctic study, Weller et al. (2015) reported the average growth and formation rates (in a size range from 3 to 25 nm) to be 0.9 nm h$^{-1}$ and 0.06 cm$^{-3}$ s$^{-1}$, respectively. These authors also concluded that due to an insufficient amount of low volatile organic compounds, the particle growth was restricted to the nucleation mode. All of these studies showed some resemblance to the results observed in our study.

The question is what mechanism drives the nucleation and what are the condensable vapors responsible for the observed particle growth in the pristine high altitude environments. Most recent studies indicate the importance of semi volatile organics (Willis et al., 2016; Burkart et al., 2017). The subsequent growth of newly formed particles was associated with organic precursors from meltwater ponds Kyrö et al. (2013), while Weller et al. (2015) speculated that low volatile organic compounds of marine origin governs the growth of newly formed particles in Antarctica. It was also shown that in a clean

environment, sufficiently high sulfuric acid concentrations ($10^7$ molecules cm$^{-3}$) can fully explain particle growth (Jokinen et al., 2018). The GRs observed in our study is somewhat similar to those from akin environments, however, they remain difficult to compare because of case-to-case variability.

    Insights on the chemical composition of nucleation mode particles and the climatic relevance of NPF can be drawn from the hygroscopicity measurements either at water vapor sub-saturation (measurements of HGF) or supersaturation (measurements

of the number of CCN). While $\kappa$ is a parameter that is independent of experimental conditions, HGF still depends on dry particle size and RH for which it was determined. Still, for the Arctic more data is available for HGF, so that we will use this parameter for comparison with literature in the following. Zhou et al. (2001) measured the HGF during the Arctic Ocean Expedition 1996. The HGF of nucleation mode particles (just after a NPF event, dry diameters of 15) was 1.38. The HGF of 35 nm particles was 1.56. After some time, the particles that grew to sizes of 50 nm were found to be less hygroscopic (HGF

of 1.05). It was suggested that these particles were produced at the sea surface and not in the free troposphere. However, the authors could not derive the composition of those nucleation mode particles. Park et al. (2014) reported HGF values of 50





nm particles during enhanced number concentration of the Aitken mode to be 1.46. Sulfate and biogenic volatile organic species were identified to contribute to the Aitken mode particle formation. Compared to our measured HGF of 15 and 20 nm particles, we can see that during Events 1 and 4 values agree reasonably well to previously measured particle hygroscopicity. The HGF of nucleation mode particles during Events 2 and 3, on the other hand, are significantly lower. The hygroscopicity of the Aitken mode particles, measured during Event 4 was almost identical to that noted by Park et al. (2014). On the other instances, for example Event 2, the HGF of the Aitken mode particles was lower (1.33 versus 1.46) than previously reported values. It clearly indicates that different condensable vapors were driving the growth of newly formed particles into sizes of 30 to 50 nm.

Based on particle hygroscopicity, formation and growth rates of positive/negative ions and neutral clusters, and offline chemical analysis, our observed NPF events represent two different cases: 1) more hygroscopic particle formation favored by negative ions, events 1 and 4 (1 June and 2 July, respectively); and 2) relatively low hygroscopicity particle formation during events 2 and 3 (18 and 26 June, respectively), suggesting the presence of condensable organics in particle growth. Further, we would like to discuss event-specific particle growth/formation rates and hygroscopicity with respect to formation mechanism and condensable vapors.

### 4.2 Indirect evidence of the composition of newly formed particles

#### 4.2.1 NPF 1 and 4

Occurrences of nucleation mode particles in the summer Arctic were associated with intrusion from higher altitudes and new particle production in upper layers of the marine boundary layer (MBL, e.g. Wiedensohler et al., 1996). It is possible that the NPF precursors can be brought from either open ocean or anthropogenic continental sources by air masses. Coupled with low condensation and coagulation sink and with plentiful global radiation it creates favorable conditions for new particles to be formed. However, in all of our observed NPF cases the particle formation started from nucleation of 1-2 nm clusters, suggesting that the NPF took place right at the sea level, rather than in upper layers of the MBL. In this study, unfortunately neither high-resolution online chemical composition of aerosol particles, nor relevant gases (e.g. $SO_2$, $O_3$) were directly measured on-board RV Polarstern. To gain some insights into the chemical composition of newly formed and slightly grown particles, as well as precursor gases, we used measured particle physico-chemical properties (e.g. hygroscopicity, growth rate, etc.) as well as satellite imagery.

It is known that Arctic phytoplankton contributes to the production of dimethyl sulfide (DMS), which is the main source of biogenic sulfur (Stefels et al., 2007; Levasseur, 2013 and reference therein). Released into the atmosphere, DMS can be involved in NPF through oxidation and creation of sulfuric acid ($H_2SO_4$) (Kulmala et al., 2001; Park et al., 2017). In a study by Nguyen et al. (2016), NPF and particle growth at a Station Nord, Greenland was found to be linked to $O_3$ most likely


through creation of hydroxyl (OH) radical and oxidation of sulfur dioxide (SO₂) and volatile organic compounds. The satellite-derived chlorophyll-a mass concentration in surface seawater, as an indicator for phytoplankton biomass (Becagli et al., 2016) can be seen in Fig. 5 (left). It is evident that during all NPF events the RV Polarstern was in a close proximity to an

area of increased biological activity in the Arctic Ocean. During Event 1, we also observed large ice-attached mats of the *Melosira arctica* (Fig. 5), which suggests the presence of DMS (Levasseur, 2013). It can also be seen from Fig. 5 that sea ice retreat is somewhat linked to increase in chlorophyll-a mass concentration in surface seawater. For example, on 26 June some ice coverage in the southern part of Prince George Land can still be visible, while on 2 July it is all gone, replaced by biological activity. This is most likely because the ice edge provides increased stability from the melt water, which facilitates

the seasonal production of phytoplankton (Conover and Huntley, 1991).

An interesting feature was observed with respect to formation rates and the number size distributions of positive and negative ions during Events 1 and 4 (Fig. 2, see also Supplementary material SP-1). Firstly, it seems that the formation of ions occurred before that of neutral particles. The peak ion formation rate was observed approx. half an hour prior the formation of neutral particles. Although not in Polar Regions, similar behavior was noticed in several other studies

suggesting the importance of ions in NPF events (Manninen et al., 2010; Jayaratne et al., 2016). The role of ions in NPF was investigated in both laboratory and field studies (e.g. Wagner et al., 2017; Jokinen et al., 2018). It was shown that ions enhance the nucleation and condensation of the vapor molecules by stabilizing the molecular clusters and/or are involved in charged cluster neutralization via recombination with oppositely charged clusters. The second interesting feature that was observed only during Events 1 and 4 was the absence of smallest (<1.6 nm) positive ions. Negatively charged ions seemed to

be involved in the particle formation more favorably than the positive ones. This was also observed in previous studies (e.g. Hirsikko et al., 2007; Asmi et al., 2010; Jokinen et al., 2018) and was associated with sulfuric-acid nucleation. Although H₂SO₄ concentrations were not determined directly, the presence of negative clusters suggests that in case of Events 1 and 4, sulfuric acid was somewhat involved in observed NPF too.

From previous studies, it was shown that $H_2SO_4$ concentration of $10^7$ molecules cm$^{-3}$ are sufficient to explain the observed

new particle GRs in coastal Antarctica (Jokinen et al., 2018). In our case, the hypothesis was tested that $H_2SO_4$ was involved in NPF Events 1 and 4 by using the look-up tables from an ion-mediated nucleation model for the $H_2SO_4$-$H_2O$ binary system (Yu, 2010). At a given temperature ($T_{Event1}$ = 268.8 K; $T_{Event4}$ = 268.4 K), relative humidity ($RH_{Event1}$ = 96.3%; $RH_{Event4}$ = 96.23%), and surface area concentration of pre-existing particles ($S_{Event1}$ = 2.9 µm$^2$ cm$^{-3}$; $S_{Event4}$ = 0.5 µm$^2$ cm$^{-3}$), and assumed ionization rate (Q = 2 ion-pairs cm$^{-3}$ s$^{-1}$) - the corresponding $H_2SO_4$ concentrations was calculated to be approx. $10^6$

molecules cm$^{-3}$. If compared to study from Antarctica (Jokinen et al., 2018) or laboratory studies by Dunne et al. (2016) from the CERN CLOUD (Cosmics Leaving Outdoor Droplets) chamber, our calculated $H_2SO_4$ concentration is 10 to 30 times lower than that from previous studies. On the other hand, the results of this study are in agreement with a study by Ehn et al. (2007), who studied the relationship between particle hygroscopicity and sulfuric acid concentration in boreal forest.





Authors reported that concentration of $H_2SO_4$, corresponding to 15 and 20 nm particle HGFs of 1.34 and 1.46, was in a range
of $10^7$ molecules $cm^{-3}$. Nevertheless, there were numerous instances when the same hygroscopic growth was also observed
at lower $H_2SO_4$ concentrations ($<10^7$). Moreover, both in the Arctic and Antarctica, $H_2SO_4$ concentrations of $10^6$ molecules
$cm^{-3}$ were associated with NPF by Croft et al. (2016) and Kyrö et al. (2013), respectively.

The fraction of the particle growth that can be explained by sulfuric acid can be found from the comparison of observed
versus predicted particle growths. From Vakkari et al. (2015), the particle growth due to sulfuric acid can be found from the
relation:

$$GR_{calc} = \frac{c_{H_2SO_4}}{A},$$ (5)

where coefficient A is equal to $1.58 \times 10^7$, $1.99 \times 10^7$, and $2.28 \times 10^7$ for particle growth in the size range from 1.5 to 3, 3 to
7, and 7 to 20 nm, respectively. Using our estimated concentration of $H_2SO_4$ we found that growth (in a size range from 1.5
to 3 nm) due to sulfuric acid alone accounts only from 4 to 10% of the observed growth during Event 4 and 1, respectively.
The contribution to particle growth in a size range from 3 to 20 nm gets even lower, 4-5%. Our values are somewhat
comparable to those observed in the Antarctica (Kyrö et al., 2013). It suggests that besides sulfuric acid, other vapors have to
be present to reach the observed particle growth. From offline chemical analysis, we see that during Event 1, ammonium and
nss-sulfate in accumulation and the Aitken mode particles were somewhat higher than campaign average (Fig. 4). Some
studies (e.g. Croft et al., 2016; Köllner et al., 2017) identified that organic species (e.g. ammonia, ammines) are linked to
particle growth in Arctic regions. If we assume that the newly formed particles were partly or fully neutralized by ammonia
(Giamarelou et al., 2016), we would expect particle hygroscopicity to be close to that of ammoniated sulfates. However, our
observed HGF of 20 and 30 nm particles during both events were lower than HGF for pure ammonium sulfate particles (1.46
versus 1.64, Asmi et al., 2010). Similar hygroscopicity of ultrafine particles (HGF = 1.38 for 15 nm particles) in the Arctic
was observed by Zhou et al. (2001). However, authors excluded the water-sulfuric acid nucleation as a source of such
particles because <50 nm particles did not appear to be composed neither of sulfuric acid nor ammonium sulfate. Compared
to a study from CLOUD (Kim et al., 2016), the measured hygroscopicity of 20 nm particles during Event 1 was closest to the
results of experiment number C (see table 1 in Kim et al., 2016), during which sulfuric acid and dimethylamine (DMA)
concentrations of $7.6 \times 10^6$ molecules $cm^{-3}$ and 23.8 ppt, respectively, resulted in HGF of 1.45 (for 15 nm particles). With that
being said, experiment E (sulfuric acid + organics produced from α-pinene ozonolysis, 420 ppt) resulted in 15 nm particles
with HGF = 1.33, which is identical to those observed during Event 4.

To conclude, one can only assume that during Events 1 and 4, the NPF was initiated by sulfuric acid. The organics of
marine-origin could have also been involved in particle growth to some extent. However, low (compared to campaign
average) organic matter concentrations, observed by offline chemical analysis, oppose to aforesaid conclusion. The





hypothesis that NPF is driven by sulfuric acid can be supported by the results of neutral cluster and ion number size

distribution and hygroscopicity measurements of nucleation mode particles.

**4.2.2 NPF 2 and 3**

Following the same line of thought as in the previous section, we investigate to what extent sulfuric acid may have been
involved in the NPF and growth during Events 2 and 3. From satellite imagery of chlorophyll-a (Fig. 5) we can see that RV
Polarstern remained in close proximity to somewhat decreased, but still present biological activity in the Arctic ocean. In

addition to that, some depletion in sea ice-cover close to Greenland, as well as increase in biological activity south of
Svalbard was also observed (Fig. 5). Thus, it is safe to say that air masses arriving at RV Polarstern were passing over
regions, which are a potential source of both DMS and organics of marine origin. Assuming a $H_2SO_4$-$H_2O$ binary system, the
$H_2SO_4$ concentrations corresponding to formation rates of those observed for Event 2 and 3 were from 15 to 50% higher,
compared to Events 1 and 4. This is mainly because during Events 2 and 3 both, the condensation sink and temperature were

higher too. Only between 1 and 3% of observed particle growth during Event 2 can be explained by $H_2SO_4$ alone. This
fraction is somewhat higher on Event 3 (6-9%). At the initial states of nucleation mode particle growth, particle
hygroscopicity on both events was rather low (HGF between 1.16 and 1.18). Such low hygroscopic particle growth, coupled
with rather rapid increase in size (Event 2, from 3 to 20 nm, GR = 4.2 nm h$^{-1}$) suggest that in these events the organics must
have played a much bigger role during initial particle growth than during Events 1 and 4. The observed particle

hygroscopicity agrees rather well with less-hygroscopic particle values reported by Zhou et al. (2001). During Event 2,
particle hygroscopicity did not change much when particles from nucleation mode grew into the Aitken mode, with HGF
remaining between 1.16 and 1.18. Only after approx. 30 hours after the new particles were created, they grew to a diameter
of 50 nm with slightly increased hygroscopicity, HGF = 1.33. Contrarily, on Event 3 the HGF of 50 nm particles (after
approx. 40 hours after the nucleation) reached values of 1.55. It is expected that with time newly formed particle

hygroscopicity will increase due to the process known as aging. From smog chamber experiments, Tritscher et al. (2011)
showed that organic aerosol photochemical aging increases the particle hygroscopicity mainly due to $O_3$ induced
condensation of organic molecules onto particles. The rate at which particle hygroscopicity parameter $\kappa$ increases can be
calculated from the change in $\kappa$ over the time period ($\Delta\kappa/\Delta t$). We found that during events 2 and 3 $\kappa$ changed with the rates
of 0.0027 and 0.0067 h$^{-1}$, respectively. These values are surprisingly close to those observed by Tritscher et al. (2011),

further supporting the evidence of organics participating in our observed particle growth.

Using our calculated formation rates (0.06 and 0.026 cm$^{-3}$ s$^{-1}$ during events 2 and 3, respectively) and sulfuric acid values
from previous studies (5 x 10$^6$ molecules cm$^{-3}$, Croft et al., 2016) as a guideline, we calculate the extremely low-volatility
organic compound concentration from the parameterization of particle formation rate as a function of sulfuric acid and EL-
VOC concentration (Riccobono et al., 2014):



$\quad J = 3.27 \times 10^{-21} cm^{-6} s^{-1} \times [H_2SO_4]^2 \times (EL-VOC).$ (6)

The resulting EL-VOC concentration for Event 2 was found to be approx. 8.0 x $10^6$ molecules cm$^{-3}$. This is 40 times higher than what is expected from monoterpenes air-sea flux in Arctic Ocean (Croft et al., 2016). On the other hand, during Event 3, the estimated concentration of EL-VOC was in pair with results published by the same authors. The question is where the EL-VOC comes from? Kyrö et al. (2013) showed that NPF can be a result of precursor vapor emission from meltwater

ponds. In Fig. 5, we can see that air masses during Event 2 is arriving from the coast of Greenland, with a pronounced sea-ice index change, indicating ice retreat. Moreover, measurements of PNSD at Villum Research Station also indicated the occurrence of NPF. However, it remains unclear if the ice and biological activity development at the coast of Greenland could have produced the organic vapors that participated in NPF observed at RV Polarstern. Yet another source of condensable organic vapor could be the aged phytoplankton blooms, presented as irregularities in chlorophyll-a spatial

distribution, at the marginal ice zone, close to research vessel.

Atmospheric particulate methanesulfonic acid (MSA) and none-sea salt sulfate (nss-sulfate) are considered to be oxidation end products of DMS, which is released as a gas during biogenic processes and indicates the formation of secondary aerosol with biogenic origin (Leck et al., 2002; Miyazaki et al., 2010). MSA was shown to be involved in nucleation mode particle growth in the Arctic by Willis et al. (2016). However, authors only hypothesized that MSA and condensable organic species

originate from marine-derived biogenic volatile organic compounds. Organic matter in Arctic submicron particles was found to be of both continental and biogenic marine origins (Kerminen et al., 1997; Chang et al., 2011). Orellana et al. (2011) showed that submicron OM can be composed of phytoplankton exudates in form of marine hydrogels. If we look at offline chemical analysis of aerosol sample, OM was found on all impactor stages, especially on the submicron particles between 0.14 and 1.2 µm. While submicron particles of the impactor samples for Episode 1 were mainly dominated by ammonium

and nss-sulfate (see Fig. 4), higher concentrations of OM (together with MSA) were found for the sampling period between 25 to 28 June. These results corresponded to the observed differences between particle hygroscopicity during events 1 and 3.

To summarize, the rapid particle growth (Event 2), and the low but steadily increasing hygroscopicity (events 2 and 3) suggest that organics must have been involved in both NPF and subsequent particle growth. Although our observed results agree with previously made conclusions that particle growth in the Arctic is largely via organic condensation (Burkart et al.,

2017), due to a lack of measurements, we cannot specify which organic species may/or may not have been involved in these processes. We also cannot exclude neither the role of iodine (Allan et al., 2015) in the initial steps of NPF, nor other pathways for initial particle growth (e.g. aminium salts; Smith et al., 2010). In the future, measurements of chemical composition of naturally charged air ions and ion clusters and low-volatile aerosol precursor gases would greatly improve our understanding about NPF processes and particle growth in the Arctic.


### 4.3 Implication for CCN abundance

In the last section of this work, the climatic relevance of the newly formed particles in the Arctic is discussed. In several studies (e.g. Allan et al., 2015; Willis et al., 2016; Burkart et al., 2017) it was reported that nucleated particles in the Arctic atmosphere rarely grow beyond the Aitken mode. It is the result of low organic vapor/precursor gas concentrations involved in NPF and subsequent growth, as well as particle lifetime (particles being scavenged by fog or precipitation, Karl et al., 2012). These findings are also comparable to those from Antarctica. Weller et al. (2015) reported that particle growth is governed by the deficit of availability of low volatility organic compounds of marine origin and made the conclusion that particles do not grow to a diameter range relevant for acting as CCN. On the other hand, some studies both from Arctic and Antarctica proved that particles do not have to grow beyond 50-60 nm in diameter to be able to act as CCN (Kyrö et al., 2013; Croft et al., 2016; Leaitch et al., 2016). This is because in the pristine Arctic environment the absence of larger particles may lower water uptake, which will increase supersaturation, enabling cloud water to condense on smaller particles (Leaitch et al., 2016).

To examine to which degree NPF may influence the CCN nuclei budget in the Arctic, we used an adiabatic non-entraining cloud parcel model (described in section 2.4). All the initial parameters and simulation results can be found in Table 3. The change in CCN number was calculated for two different updraft wind velocities, 0.1 and 3.2 m s$^{-1}$, representing the 75$^{th}$ percentile and maximum value, respectively. The measurements of vertical wind velocity was performed during the ice-drift station, as described by Egerer et al. (2019), and can also be found online (https://doi.pangaea.de/10.1594/PANGAEA.899803). We define the CCN number concentration ($N_{CCN}$) increase due to particles created in the nucleation process as:

$$Increase\ of\ N_{CCN} = \frac{N_{CCN,bp} + N_{CCN,NPF}}{N_{CCN,bp}}, \tag{7}$$

where $N_{CCN,bp}$ is the number concentration of CCN resulting from background aerosol particles (particle diameter >100 nm) and $N_{CCN,NPF}$ is the CCN number concentration resulting from the particles created in NPF event. It can be seen that for most of the cases (when RH>90%), the CCN number concentrations increased by a factor of 2 to 5 (at upward wind velocities of 0.1 m s$^{-1}$) and 4 to 32 (at upward wind velocities of 3.2 m s$^{-1}$). Although the activated number fraction in a size range from 15 to 20 nm was rather low (1.5 - 4%), the high number of nucleation mode particles resulted in a noticeable increase of total CCN. The CCN fraction was higher (30 - 50%) when 3.2 m s$^{-1}$ updraft wind speed was assumed. For the Aitken mode particles, CCN fraction was approx. 12 and 80% for updraft wind speeds of 0.1 and 3.2 m s$^{-1}$, respectively. In some cases, the particles did not activate to CCN. This is because activation supersaturation was not reached during the parcel updraft. The maximum supersaturation achieved with an updraft velocity of 0.1 m s$^{-1}$ was 0.17%. The updraft velocity of 3.2 m s$^{-1}$ would represent although rare, however, not unlikely situation when supersaturations of 0.9% can be reached. It can be





anticipated that an even higher fraction of CCN may result from nucleation mode particles when higher supersaturation
values are reached. Measurements of CCN number concentration on-board RV Polarstern corroborate the results obtained by
our modelling efforts, which all are in good agreement with previous works. For example, Croft et al. (2016) reported
maximum supersaturation in the Arctic region of 0.15–0.25% for the updraft speed of 0.1 m s$^{-1}$. From a comprehensive study
on the ultrafine particle effects on liquid clouds in the clean summertime Arctic, Leaitch et al. (2016) determined the

supersaturation for low and high altitude clouds to be approx. 0.3 and 0.6%, respectively. In the Arctic environment with the
lack of aerosol particles upon which clouds may form, even a small increase in aerosol loading can lead to cloud formation
and thus influence the ice-covered Arctic surface (Mauritsen et al., 2011). From our results, we conclude that NPF in the
Arctic can play a significant role in determining the future changes in this pristine and remote environment.

**5 Summary and conclusion**

Aerosol particle physico-chemical properties were determined in the summer Arctic on-board research vessel (RV)
Polarstern from 26 May to 16 July 2017 as a part of the PASCAL/SiPCA campaign. Here, regional NPF events are analyzed
and put into prospective of producing the CCN. From the measurements of neutral cluster and air ion number size
distributions, it can be conclude that new particles were formed within the marine boundary layer and not mixed down from
aloft. Therefore, the majority of particles in a size range up to 50 nm in diameter can be related to secondary aerosol

production rather than primary emissions. Two different types of NPF were distinguished: a) NPF favored by negative ions,
and more-hygroscopic nucleation mode particles; and b) NPF with subsequent rapid growth (Event 2), resulting in less-
hygroscopic particles. From analysis of particle formation and growth rates, as well as the hygroscopicity of slightly grown
particles, it seems that sulfuric acid-water ion-mediated nucleation is an acceptable mechanism explaining the observed NPF
during events 1 and 4. Meanwhile, low particle hygroscopicity and rapid growth suggests that condensable organics were

somewhat involved in particle growth during events 2 and 3. Although the imagery from satellite confirms the biological
activity as a possible source of marine sulfur and organics, however, due to the lack of appropriate measurements we cannot
provide quantitative information to what extent these precursor gases may have been involved in the observed particle
formation and growth. For the same matter, we also cannot exclude other species (e.g. iodine) participating in NPF. To
answer these questions, high temporal resolution measurements of nucleation and the Aitken mode particle chemical

composition after the NPF is necessary, which remains a topic for the future research.

After the nucleation, in 12 to 56 hours newly formed particles grew to the Aitken mode sizes (approx. 30-50 nm). We have
traced particle growth and measured particle hygroscopicity for dry diameters of 15, 20, 30, and 50 nanometers. Here, one of
our main objectives was to test whether particles created in the Arctic marine boundary layer can act as CCN. To accomplish
this task we have used a zero-dimensional, adiabatic cloud parcel model. Measured particle physico-chemical properties and

ambient information (relative humidity, pressure, temperature) were used to simulate particle population activation to cloud
droplets at two different updraft velocities of 0.1 and 3.2 m s$^{-1}$. Simulation results showed that although the activated fraction





of nucleation mode particles were below 5% at an updraft wind velocity of 0.1 m s$^{-1}$, background CCN number concentration increased by up to a factor of 5. The Aitken mode particle activation was somewhat higher, approx. 12%. Such increase in CCN number concentrations was also confirmed by direct measurements for supersaturations from 0.1 to 1% on-board RV

Polarstern. Our findings support previous observations suggesting that in pristine Arctic environment particles do not have to grow to sizes above 50 nm to act as CCN. We conclude that in a changing Arctic, NPF can be an important source of CCN. New particle formation and the Aitken mode particle ability to become CCN requires more in depth studies with the focus on mechanisms of NPF, chemical composition of the precursor gases and condensable vapors, as well as the identification of their sources and impact on Arctic clouds.

**Data availability**

Processed and raw data available on request from corresponding author.

**Author contribution**

SK – operated aerosol instrumentation on-board RV Polarstern, evaluated data, and wrote the manuscript.

TV – operated aerosol instrumentation on-board RV Polarstern and contributed to manuscript writing.

PP, JL, and MK – contributed to NAIS data evaluation, discussion and manuscript writing.

DR – contributed to the simulation of CCN.

HW – contributed writing the manuscript.

SZ and MP – collected samples for chemical analysis. Contributed writing the manuscript.

MH – operated the CCNC on-board RV Polarstern and evaluated CCNC data.

XG and AW – operated CCNC on-board RV Polarstern.

SH – calibrated the CCNC prior measurement campaign.

FS, HH, and AW – participated in fund raising for the measurement campaign.

**Competing interests**

The authors declare that they have no conflict of interest.



**Special issue statement**

This article is part of the special issue "Arctic mixed-phase clouds as studied during the ACLOUD/PASCAL campaigns in the framework of (AC)³ (ACP/AMT inter-journal SI)". It is not associated with a conference.

**Acknowledgements**

We gratefully acknowledge the funding by the Deutsche Forschungsgemeinschaft (DFG, German Research Foundation) – Projektnummer 268020496 – TRR 172, within the Transregional Collaborative Research Center "ArctiC Amplification: Climate Relevant Atmospheric and SurfaCe Processes, and Feedback Mechanisms (AC)³, as well as funding of the Polarstern cruise PS106 (expedition grant number AWI-PS-106-00) by AWI. Authors would also like to acknowledge a number of people, who were involved in this work. We acknowledge the discussions and support (H2SO4-H2O nucleation look-up tables) by Fangqun Yu (UAlbany). We also thank Sebastian Ehrhart (MPIC), Joachim Curtius (IAU), Steffen Münch (ETHZ), and Andreas Kürten (IAU) for the discussions concerning sulfuric acid-water nucleation. Ella-Maria Duplissy, Veli-Matti Kerminen, Jenni Kontkanen, Stephany N. Buenrostro Mazon from Helsinki University for their time, valuable suggestions, and discussions. Ulrike Egerer for providing the updraft wind velocities during the ice-drift station. Hannes Griesche, Ronny Engelmann and Martin Radenz for providing ship-based remote sensing data to characterize the cloud situations during the selected events. Peter Gege (DLR), Svenja Kohnemann (UniTrier), and Marcel Nicolaus (AWI) for sharing the ship-deck photos. Andreas Macke (TROPOS) and Hauke Flores (AWI), Chief Scientists of PS106 cruise, for the attitude and phenomenal attention to all our requests regarding scientific activities on-board RV Polarstern and on the ice. And finally, the RV Polarstern crew, staff members, numerous scientists, and Polar Bear guards and watchers, who made the expedition not only exceptional, but also a safe experience. Villum Research Station, Robert Lange, Andreas Massling, Henrik Skov, and Niels Bohse Hendriksen are acknowledged for providing PNSD data. We acknowledge Hartmut and Andrea Haudek for building the conditioning system for both aerosol inlet and the Berner impactor for these Arctic environmental conditions. Maik Merkel and Rene Rabe was a huge technical support for setting up the measurement container and Berner impactors. Susanne Fuchs performed the ion chromatography analysis and Anke Rödger the OC/EC thermographic analysis. We also acknowledge the use of imagery from the NASA Worldview application (https://worldview.earthdata.nasa.gov), part of the NASA Earth Observing System Data and Information System (EOSDIS). Also, this study has been conducted using E.U. Copernicus Marine Service Information (Arctic Chlorophyll Concentration from Satellite observations (daily average) Reprocessed L3 (ESA-CCI).





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

**Figure 1: Cruise track and particle number concentration (integrated in a size range from 10 to 800 nm) during PASCAL and SiPCA expeditions. The days, which were picked to analyze NPF events and subsequent particle growth, are indicated with square boxes. Backward air mass trajectories (72 hours) were calculated using HYSPLIT (Draxler and Rolph, 2012), and are shown by solid (200 m a.s.l) and dotted (2000 m a.s.l) lines corresponding each NPF event. Ice-drift is shown in the insert. Thin blue and black lines are the observed ice-edge for June and July, 2017, respectively (Fetterer et al., 2002).**



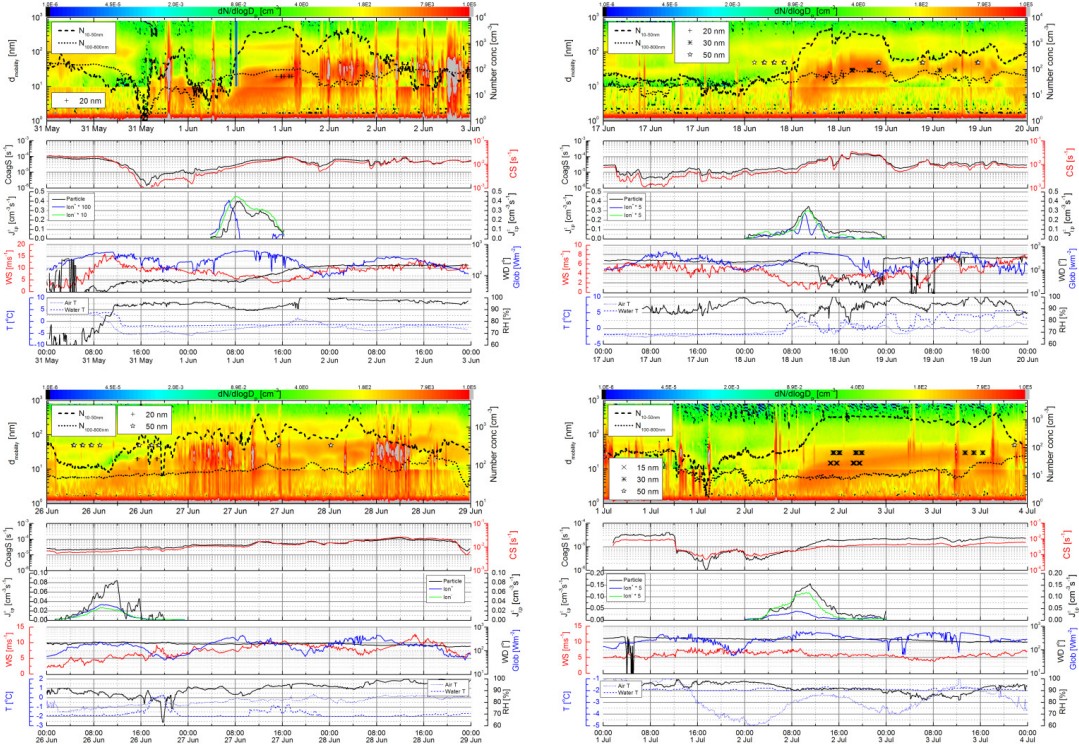

**Figure 2: The NPF events observed during RV Polarstern cruise PS106. The PNSDs from NAIS (negative polarity) and MPSS are shown as contour plots. The color scale represents particle number concentration as *dN/dlogD_p*. Inside the contour plots, particle number concentration, integrated between two size ranges (10 to 50 nm and 100 to 800 nm) is shown with dashed and dotted black lines. The presence of corona charger ions (<2 nm, Manninen et al., 2011) can also be seen in NAIS data. This artefact was excluded from data analysis. Coagulation and condensation sinks, meteorological parameters (wind speed and direction, global radiation, temperature, and relative humidity), and formation rates (*J*) for each NPF event are shown in the panels below the contour plots. Note: sample contamination by ship exhaust was removed from data analysis, however, for better representation of particle growth, the contour plots include all the data (contamination not removed).**

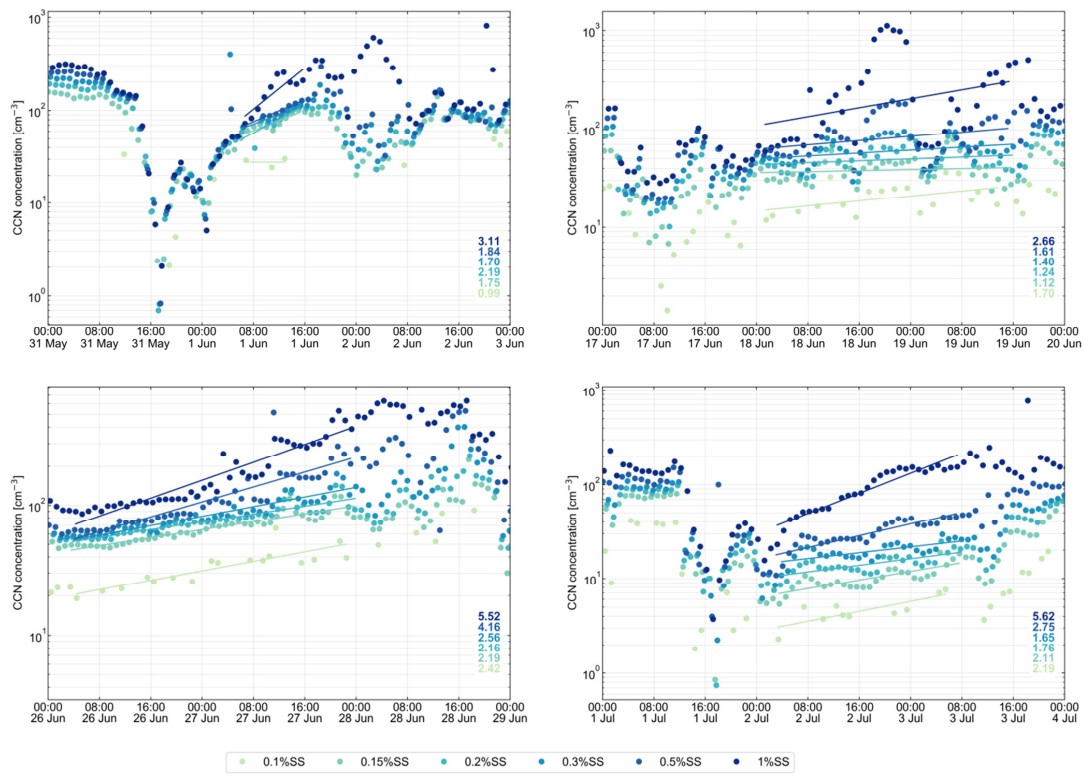

**Figure 3: The CCN number concentration measured during NPF events (1 to 4). The lines and corresponding values show the increase in CCN concentrations (prior NPF vs. particles have grown to the Aitken mode).**






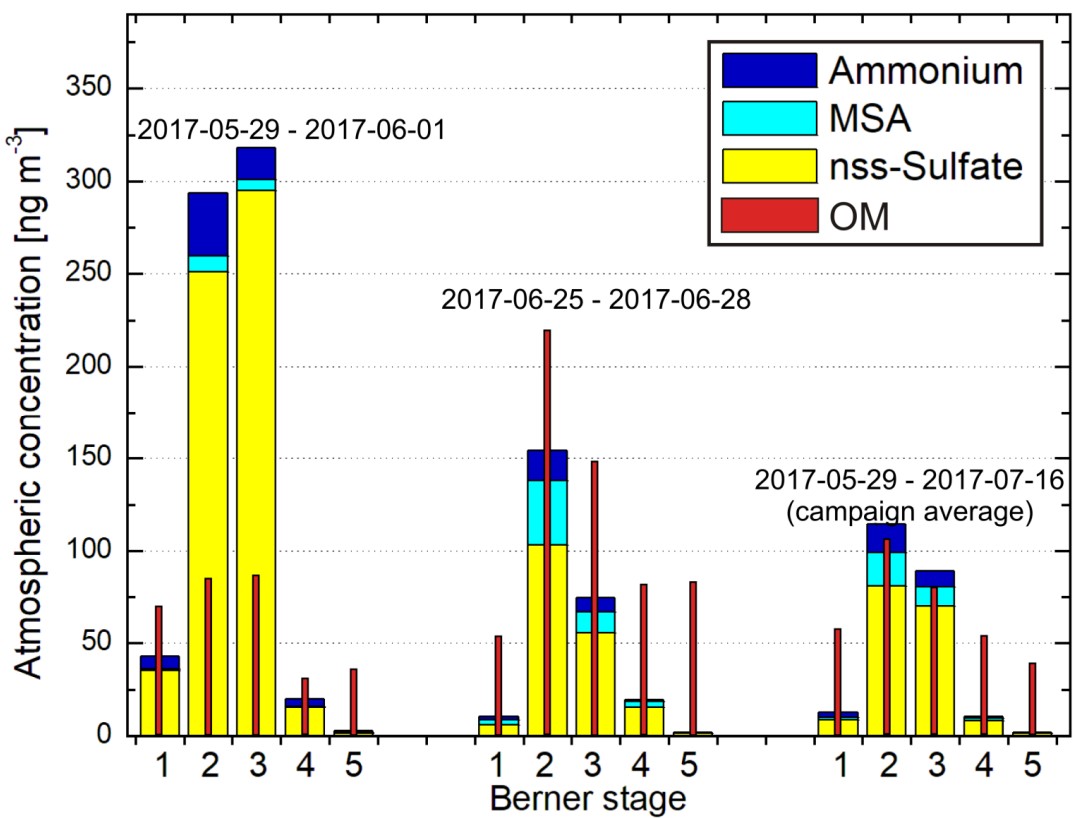

**Figure 4: Size-resolved atmospheric concentrations for ammonium, MSA, nss-sulfate, and OM for two sampling periods and the whole campaign average.**





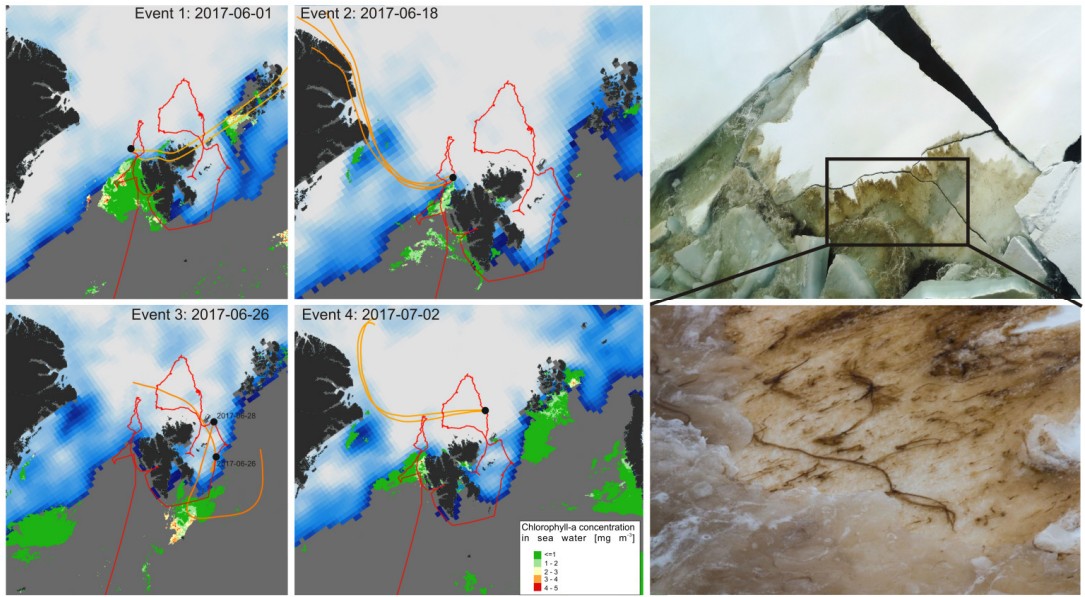

**Figure 5: Sea ice concentration (white - 100%, dark blue - 1%; from NASA Worldview; Maslanik and Stroeve, 1999) and chlorophyll-a surface concentration (taken from http://marine.copernicus.eu, accessed 29 April, 2019) during NPF events (left). On the right - the ice alga and diatom *Melosira arctica* (right) observed from the ship deck during the NPF Event 1. RV Polarstern track and location during NPF event is indicated in red line and black circle, respectively. Orange lines mark 72 hour backward air mass trajectory at 200 m.a.s.l.**


**Table 1: Calculated parameters for observed NPF events during RV Polarstern cruise 106. The GR is obtained from NAIS size spectrum using the methods proposed by Kulmala et al. (2012) and Pichelstorfer et al (2018). $J$ is the formation rate of 1.6-nm sized positive/negative ion clusters and 3-nm sized particles. Please note that in some instances the size range for GR and $J$ calculations is different (due to measured PNSD). Nevertheless, we calculated both parameters from the smallest possible particle/ion size range. Value after "±" shows standard deviation.**

| Event | Date (of 2017) | Ship position | GR [nm h$^{-1}$] (size range) | | |
|---|---|---|---|---|---|
| | | | Particle (3-7 nm) | Ion$^{+}$ | Ion$^{-}$ |
| 1. | 06-01 | 80.4N; 7.2S | 1.2 ± 0.05 | 1.43 (1.6-4 nm) | 0.66 (1.6-3 nm) |
| 2. | 06-18 | 80.2N; 10.7S | 4.25 ± 0.89 | 3.30 (4-9 nm) | 2.90 (1.6-3 nm) |
| 3. | 06-26 | 78.4N; 33.4S | 0.62 ± 0.16 | 2.16 (2-6 nm) | 1.22 (1.6-3 nm) |
| 4. | 07-02 | 81.6N; 33.3S | 0.88 ± 0.01 | 3.43 (1.6-4 nm) | 1.49 (2-3 nm) |
| | | | $J$ [cm$^{-3}$ s$^{-1}$] | | |
| | | | Particle ($J_3$) | Ion$^{+}$ | Ion$^{-}$ |
| 1. | 06-01 | 80.4N; 7.2S | 0.39±0.05 | 0.004 ($J_{1.6-}$) | 0.045 ($J_{1.6-}$) |
| 2. | 06-18 | 80.2N; 10.7S | 0.35±0.03 | 0.054 ($J_{4-}$) | 0.060 ($J_{1.6-}$) |
| 3. | 06-26 | 78.4N; 33.4S | 0.08±0.01 | 0.033 ($J_{2-}$) | 0.026 ($J_{1.6-}$) |
| 4. | 07-02 | 81.6N; 33.3S | 0.15±0.01 | 0.007 ($J_{1.6-}$) | 0.023 ($J_{2-}$) |

**Table 2: Hygroscopic growth factor (at 90% RH) and hygroscopicity parameter $\kappa$ during NPF events. Here, Time of scans - a time window during which hygroscopicity distributions were measured; $t_J$ - approx. time between the observed formation rate maximum and the measurements of HGF. In other words, $t_J$ indicates how long before/after the NPF events the HGF was measured. For example, if $t_J = 7$, the HGF was measured 7 hours after the maximum in $J$. Negative $t_J$ indicates the measurements of HGF prior NPF event; $d_0$ - selected diameter of dry particles; $N_{scans}$ - number of scans; sd - standard deviation.**

| Time of scans | $t_J$ [h] | $d_0$ [nm] | HGF±sd | $\kappa$±sd | $N_{scans}$ |
|---|---|---|---|---|---|
| *Event 1* | | | | | |
| 06.01 15:00 - 17:41 | 7.0 | 20 | 1.46±0.02 | 0.41±0.02 | 11 |
| *Event 2* | | | | | |
| 06.18 12:14 - 16:52 | 1.6 | 20 | 1.17±0.02 | 0.13±0.00 | 10 |
| 06.18 18:11 - 21:21 | 7.6 | 30 | 1.17±0.02 | 0.11±0.00 | 6 |
| 06.19 09:06 - 11:44 | 22.5 | 30 | 1.43±0.05 | 0.36±0.08 | 3 |
| 06.18 01:39 - 06:45 | -8.9 | 50 | 1.36±0.08 | 0.24±0.07 | 10 |
| 06.18 22:40 - 22:50 | 12.1 | 50 | 1.26±0.04 | 0.16±0.04 | 3 |
| 06.19 06:07 - 06:18 | 19.5 | 50 | 1.25±0.01 | 0.16±0.00 | 3 |
| 06.19 15:31 - 15:42 | 29.0 | 50 | 1.33±0.01 | 0.21±0.00 | 3 |
| *Event 3* | | | | | |
| 06.26 15:18 - 18:47 | 6.1 | 20 | 1.16±0.01 | 0.12±0.02 | 6 |
| 06.26 04:29 - 19:04 | -4.8 | 50 | 1.28±0.03 | 0.16±0.03 | 20 |
| 06.27 15:21 - 15:32 | 30.1 | 50 | 1.48±0.09 | 0.33±0.06 | 4 |
| 06.28 00:12 - 00:17 | 39.1 | 50 | 1.55±0.01 | 0.38±0.00 | 2 |
| *Event 4* | | | | | |
| 07.02 14:27 - 19:38 | 4.0 | 15 | 1.34±0.01 | 0.33±0.02 | 18 |
| 07.02 14:56 - 19:58 | 4.5 | 30 | 1.46±0.02 | 0.35±0.01 | 16 |
| 07.03 13:20 - 16:30 | 26.9 | 30 | 1.53±0.04 | 0.42±0.03 | 9 |
| 07.03 21:43 - 21:54 | 35.3 | 50 | 1.44±0.02 | 0.34±0.04 | 2 |





**Table 3: Input parameters and the results from parcel model (Rothenberg and Wang, 2016). Here, $P$ – pressure (Pascal), $T$ – temperature (Kelvin), RH – relative humidity (%), GMD – geometric mean diameter of two modes fitted to PNSD (in nanometers); $N$ – number concentration of particles in the mode (in particles per cubic centimeter), $\kappa$ - hygroscopicity parameter kappa (derived for particle sizes indicated in bracket), $\sigma$ - is the shape parameter (standard deviation of the log of the distribution), $N_{CCN,0.1}$ and $N_{CCN,3.2}$ is the number concentration of CCN at two different vertical wind velocities, 0.1 and 3.2 m s$^{-1}$. Note: $\kappa$ for specific GMDs was adopted from the nearest value of measured 15, 20, 30, 50, and 150 nm particle hygroscopicity. For example, hygroscopicity of 20 nm particles was used as an input value for GMD of 16 nm mode particles.**

| Time | $P$ [Pa] | $T$ [K] | RH [%] | GMD [nm] | $N$ [cm$^{-3}$] | $\kappa$ | $\sigma$ | $N_{CCN,0.1}$ [cm$^{-3}$] | $N_{CCN,3.2}$ [cm$^{-3}$] |
|---|---|---|---|---|---|---|---|---|---|
| 2017-06-01 12:00-16:00 | 102715 | 271.5 | 92.0 | 16 | 3411 | 0.41(20) | 1.4 | 0 | 1058 |
| | | | | 144 | 112 | 0.52(150) | 1.8 | 100 | 112 |
| 2017-06-18 12:00-16:00 | 100868 | 272.7 | 91.0 | 23 | 2574 | 0.13(20) | 2.2 | 104 | 900 |
| | | | | 194 | 33 | 0.28(150) | 1.7 | 32 | 33 |
| 2017-06-18 20:00-21:00 | 100839 | 273.6 | 94.6 | 38 | 2614 | 0.11 (30) | 1.9 | 156 | 1404 |
| | | | | 184 | 44 | 0.25(150) | 1.8 | 41 | 44 |
| 2017-06-19 08:00-12:00 | 100887 | 273.3 | 94.2 | 33 | 415 | 0.36(30) | 1.9 | 43 | 327 |
| | | | | 150 | 66 | 0.25(150) | 2.7 | 47 | 64 |
| 2017-06-19 15:00-17:00 | 100958 | 272.7 | 97.3 | 44 | 491 | 0.21(50) | 1.7 | 86 | 435 |
| | | | | 162 | 31 | 0.25(150) | 2.0 | 28 | 31 |
| 2017-06-26 04:00-12:00 | 100830 | 272.0 | 87.8 | 40 | 69 | 0.16(50) | 1.8 | 0 | 0 |
| | | | | 143 | 58 | 0.37(150) | 2.0 | 0 | 0 |
| 2017-06-26 15:30-16:30 | 100772 | 272.4 | 85.0 | 13 | 588 | 0.12(20) | 1.8 | 0 | 0 |
| | | | | 151 | 66 | 0.37(150) | 2.2 | 0 | 0 |
| 2017-06-28 00:00-01:00 | 100422 | 272.9 | 93.8 | 43 | 503 | 0.38(50) | 1.8 | 55 | 448 |
| | | | | 164 | 89 | 0.39(150) | 2.2 | 69 | 88 |
| 2017-07-02 16:00-20:00 | 101417 | 270.4 | 91.7 | 13 | 1121 | 0.33(15) | 1.8 | 17 | 344 |
| | | | | 112 | 20 | 0.56(150) | 2.1 | 18 | 20 |
| 2017-07-03 08:00-10:00 | 101382 | 271.4 | 84.4 | 25 | 814 | 0.42(30) | 1.9 | 0 | 0 |
| | | | | 101 | 27 | 0.65(150) | 3.0 | 0 | 0 |
| 2017-07-03 21:00-23:00 | 101039 | 270.2 | 93.9 | 35 | 207 | 0.34(50) | 2.0 | 40 | 178 |
| | | | | 125 | 55 | 0.65(150) | 1.9 | 50 | 55 |