# Peer review of "New particle formation and its effect on CCN abundance in the summer Arctic: a case study in Fram Strait and Barents Sea"

_Atmospheric Chemistry and Physics, 2019_

## Referee Comment (RC1) · Anonymous Referee #1 · 13 Aug 2019

General comments:

The main objective of this field study is to assess the impact of new particle formation (NPF) and secondary aerosol on the cloud condensation nuclei (CCN) budget in the summertime Arctic troposphere. To this end, a physico-chemical characterization of newly formed particles, their nucleation process and growth constituted the scaffolding of this work. The authors succeeded to run a formidable state of the art equipment onboard RV Polarstern during an Arctic research cruise. The presented results and conclusions are predicated on an in-depth and sound data evaluation. The authors clearly state assumptions and shortcomings (e.g. the lack of gaseous H2SO4 or in

situ organic carbon measurements, which would have been beneficial to constrain nucleation and growth mechanism). Nevertheless, the presented outcome of this work is a considerable progress in understanding aerosol-cloud interaction and its climatic consequences during Arctic summertime. In my opinion, the authors organized their manuscript straightforward and their conclusions are widely comprehensible. Without doubt, the topic addresses the scientific scope of ACP and I recommend a final publication after some minor revisions I specified below.

1. Chapter 2.1.1: I think you should better move the whole section to the corresponding places in the "Results" chapter, because it is reasonable to present this auxiliary information along with the described NPF events. In the present form, part of the information given now in chapter 2.1.1 are just repeated in chapter 3.

2. Chapter 2.2.3, line 204 & 205: Please briefly specify in which way you determine particle hygroscopicity (I guess kappa is derived from CCNC data?).

3. Page 10, lines 283 to 287: Contamination control: Did you entirely rely on the CNC data or did you also include relative wind direction and velocity from the ships weather station?

4. Page 10, lines 289 to 291: Data from Zeppelin Mountain Observatory: If these data are significant for your evaluation, I miss a more detailed discussion along with your results.

5. Page 18, lines 535 to 540: Estimating gaseous H2SO4 concentrations is crucial in order to describe nucleation and growth mechanism. If I understand aright, you used the tables and the given interpolation method presented by Yu (2010), where the basic input parameter is the observed nucleation rate (and not the growth rate)? Please clarify. In addition, you assume that because of the measured hygroscopicity parameter, only the binary system H2SO4-H2O is in the running and particularly not the tenary H2SO4-NH3-H2O system. In my view, it may still be worthwhile to envisage the latter option by comparison (see Napari et al., 2002).

6. Page 19, line 559: Sorry for nagging, but being a chemist, I have to note that ammonia is not an organic species (though it could be biogenic, if you mean this).

7. Page 22, line 655 and eq. (7): The CCN increase due to NPF is defined as the difference between the CCN formed under background aerosol with diameter >100 nm and the CCN resulting from NPF. I do not understand why you used a cut-off of 100 nm defining background conditions. First, the model handles a bi-modal size distribution with geometric mean diameter and geometric standard deviation as input. Secondly, during all NPF events to be considered (referring to Fig. 2), particles did not grow above 50 nm. Thus, if at all, it would be reasonable defining a cut-off at around 50 nm instead of 100 nm for the background case. Please clarify!

8. Figure 2: This series of figures show a wealth of information that I could not really decipher in the present printed version. A magnification of the on-line version is necessary, but the resolution of the figures are somewhat poor. I would therefore recommend remedying this point.

9. Figure 4: Fine, but what is about sea salt ($Na+$ and/or $Cl-$) data? These results are equally of some interest. For instance, a correlation with OC could give a hint whether part of the OC is primary aerosol that may be generated in conjunction with sea spray (just a mere suggestion for my part).

Typos (maybe not complete!):

10. Page 16, line 453: It should be 5.4 nm h-1 (not 5.4 h-1).

11. Page 23, line 678: . . .it can be concluded (not conclude).

Reference:

Napari, I., Noppel, M., Vehkamäki, H., and Kulmala, M.: Parametrization of tenary nucleation rates for H2SO4-NH3-H2O vapors, J. Geophys. Res.: 107(D19), 4381, doi: 10.1029/2002JD002132, 2002.

---

## Referee Comment (RC2) · Anonymous Referee #2 · 19 Aug 2019

This manuscript describes an interesting and valuable set of measurements made aboard Polarstern in the region near Svalbard, during May to July 2017. The authors report measurements of neutral and charged clusters  particle size (together covering a size range on <1 - 800nm), particle hygroscopicity and cloud condensation nuclei. Measurements of nucleation mode particle size and charged/neutral cluster abundance have been rarely made in Arctic regions, making this a unique and valuable data set that indirectly furthers our knowledge on the identity of species responsible for Arctic nucleation and initial growth of particles to Aitken mode sizes. The manuscript is well written and largely well organized. The following comments are intended to improve an already very good manuscript.

[Figure]

**General Comments:**

(1) L65-73: This is an important section of the introduction, and at present is missing several of the relatively small amount of studies that exist. Since so few data sets exist, it is reasonable to cite them even if they are not all discussed in detail. Some that should be included are listed here: Freud et al, doi:10.5194/acp-17-8101-2017; Nguyen et al, doi: 10.5194/acp-16-11319-2016; Dall'Osto et al, doi: 10.5194/acp-19-7377-2019; Burkart et al, doi: 10.5194/acp-17-5515-2017 and 10.1002/2017gl075671; Collins et al., doi: 10.5194/acp-17-13119-2017; Leaitch et al., doi: 10.12952/journal.elementa.000017; Tremblay et al., doi: doi.org/10.5194/acp-19-5589-2019; These and others were recently reviewed in doi: 10.1029/2018rg000602.

(2) The discussion of ship deck observations in Section 2.1.1 may serve the reader better if they were incorporated into the larger discussion of NPF events in Section 3.1

(3) A detailed description of how pollution influence from the ship was reproducibly removed from the data set needs to be included in the Methods section. Was the influence quantified with a specific measurement (e.g., rapid variability in particle concentrations in a certain size range?)? Or, was a specific wind sector removed from the data? "Abrupt and short increases in particle number concentration recorded by a CPC" as cited in L286 as the method for filtering ship stack pollution; this should be elaborated in the methods section and detail is needed on how the authors determined that "most of the cases when the particle number increased tenfold" could be attributed to NPF. I strongly discourage the authors from including periods of ship pollution in Figure 2 (noted in L374-375). I disagree that "better representation of particle growth" (L1031-1032) warrants this.

(4) Further to comment (3) above, for the offline chemical analysis the sampling time was 72-144 hours (L223); how was sampling pollution from the ship minimized or avoided (e.g., by a wind switch)? The presence or absence of any such precautions should be stated, and if they were absent the possible implications should be

discussed.

(5) A slightly more descriptive title might be helpful. For example, "during PS106 cruise" could be replaced by a few words describing the region of measurements. This would be helpful for readers not familiar with the region covered by the cruise.

**Specific Comments:**

(1) L44: This statement might be best attributed to a paper such as Croft et al., Nature Communications, 2016, that makes a more direct connection to radiative forcing

(2) L60-62: This statement is more attributable to Willis et al., Reviews of Geophysics, 2018 (https://agupubs.onlinelibrary.wiley.com/doi/full/10.1029/2018RG000602) and Abbatt et al., ACP, 2019 (https://www.atmos-chem-phys.net/19/2527/2019/acp-19-2527-2019.html)

(3) L63: Dall'Osto et al., Scientific Reports, 2018 corroborates these results for different regions and multiple stations (https://www.nature.com/articles/s41598-017-17343-9 and https://www.nature.com/articles/s41598-018-24426-8)

(4) L206: I assume that the CCN measurements were made on poly disperse aerosol, but this is not explicitly stated.

(5) L218-219: A large fraction of the organic aerosol may be semivolatile (e.g., Burkart et al, GRL, 2017). How might the semivolatile fraction be impacted by both heating of the inlet and long sampling times? Possibly biases introduced into these measurements should be discussed.

(6) L390-391: This is a very interesting observation and is in agreement with Tremblay et al., 2018 (doi above), could these data be included in the Supplement?

(7) L425-429: Burkart et al., 2017 (doi above) came to similar conclusions

(8) L431-441 and Figure 4: Rather than referring to the stage number here I suggest the authors refer to the corresponding size range.

(9) Related to comment (8), are any accumulation mode particle present during growth events? Do these larger modes also grow during NPF? Is sea salt present in accumulation mode sizes? Collins et al., 2017 (doi above) and Burkart et al., 2017 (doi above) observed growth of multiple modes at different rates showing that the species responsible for growing the larger mode was more semi volatile. If this is the case can the composition of the larger modes be connected to what is growing the smallest particles?

(10) L444-446: This sentence described a very unique aspect of this study which could be highlighted more, for example in the abstract or at the end of the introduction.

(11) L450-460: Comparison to other Arctic studies that report growth rates might be more appropriate here, though I do not dispute the value of comparing to Antarctic studies. Collins et al., 2017 (doi above) report growth rates for events in the Canadian Arctic during two summers. Nieminen et al., ACP 2018 (doi: 10.5194/acp-18-14737-2018) include growth rates from Alert. Available observations were reviewed by Willis et al., 2018 (doi above)

(12) L567-569: Describing the experiments (i.e., the information in brackets) might be more useful than using experiment numbers for those readers not closely familiar with the CLOUD experiments

(13) L572-573: If the measured Aitken mode particles were all organic, how much OM mass would you expect and how does that compare with the measured masses? Also, if the material is largely semivolatile how much do you expect that losses during sampling would impact this assessment?

(14) L619-620: The authors of Willis et al., 2016 provide reasonable evidence for a marine source. For example, that the organic and MSA driven growth was only observed in a shallow marine inversion layer and not aloft. Burkart et al., ACP, 2017 corroborate a marine source of NPF precursors.

(15) L634-636: The cited observations, as well as Collins et al., 2017, do demonstrate some growth into sizes above 50nm. The same work also demonstrates frequent simultaneous growth of multiple modes, and a resulting strong impact on CCN.

(16): L660-670: This is a useful analysis, and I don't suggest that the authors make substantial changes here. However, I do wonder if updraft the most appropriate way to assess this for summer Arctic low level clouds directly impacted by marine sources? Advection from warmer to colder surfaces in a shallow boundary layer might be another mechanism for CCN to active in low altitude clouds, suggested by Leaitch et al., 2016. Marine influence was significantly less for the upper level clouds observed in Leaitch 2016 (see Bozem et al., ACPD, 2019 doi: 10.5194/acp-2019-70)

(17): L706: Will these data be made publicly available in the future?

**Technical Corrections:**

(1) L144: "0 am" typo? (2) L471: "is" to "are" (3) Supplement Figure S1: typo in the y-axis label? [m] to [nm]?

---

## Author Comment (AC1) · 27 Sep 2019

Dear referee #1,

Please see the attached documents, which include the point-by-point answer to the comments, updated manuscript (with marked changes), as well as the supplementary material.

Please note that the version of manuscript already includes the comments from both referee #1 and #2.

Sincerely,

Simonas

p.s. All documents are uploaded as supplement zip file.

Please also note the supplement to this comment:
https://www.atmos-chem-phys-discuss.net/acp-2019-600/acp-2019-600-AC1-
supplement.zip

---

## Author Comment (AC2) · 27 Sep 2019

Dear referee #2,

Please see the attached documents, which include the point-by-point answer to the comments, updated manuscript (with marked changes), as well as the supplementary material.

Please note that the version of manuscript already includes the comments from both referee #1 and #2.

Sincerely,

[Figure]

Simonas

p.s. All review documents are included in the attached supplementary zip file.

Please also note the supplement to this comment:
https://www.atmos-chem-phys-discuss.net/acp-2019-600/acp-2019-600-AC2-
supplement.zip

---

## Author Response (AR1)

**Response to comments of referee #1**

**General Comments**

The main objective of this field study is to assess the impact of new particle formation (NPF) and secondary aerosol on the cloud condensation nuclei (CCN) budget in the summertime Arctic troposphere. To this end, a physico-chemical characterization of newly formed particles, their nucleation process and growth constituted the scaffolding of this work. The authors succeeded to run a formidable state of the art equipment onboard RV Polarstern during an Arctic research cruise. The presented results and conclusions are predicated on an in-depth and sound data evaluation. The authors clearly state assumptions and shortcomings (e.g. the lack of gaseous H2SO4 or in situ organic carbon measurements, which would have been beneficial to constrain nucleation and growth mechanism). Nevertheless, the presented outcome of this work is a considerable progress in understanding aerosol-cloud interaction and its climatic consequences during Arctic summertime. In my opinion, the authors organized their manuscript straightforward and their conclusions are widely comprehensible. Without doubt, the topic addresses the scientific scope of ACP and I recommend a final publication after some minor revisions I specified below.

**REPLY:**
We are contented that referee #1 sees the value of such studies. We also sincerely thank to referee #1 for his/her time, comments, and suggestions, which will indeed improve the manuscript. In the following, we would like to address the detailed comments step-by-step.

**DETAILED COMMENTS**

1. Chapter 2.1.1: I think you should better move the whole section to the corresponding places in the "Results" chapter, because it is reasonable to present this auxiliary information along with the described NPF events. In the present form, part of the information given now in chapter 2.1.1 are just repeated in chapter 3.

**REPLY:**
We agree with the referee #1 that information presented in chapter 2.1.1 can be moved to chapter 3. We also realized that part of the auxiliary information is truly repeated in chapter 3. Thus, we reviewed the mentioned chapters, dissolved chapter 2.1.1, and updated chapter 3 accordingly. Thank you for the insightful suggestion.

   We have also noticed that chapter 3.1.3 shall indicate 26 June, and not 28. Changes are highlighted in yellow.

**3.1.1 NPF 1: 1 June**

[revised manuscript text omitted]

2. Chapter 2.2.3, line 204 & 205: Please briefly specify in which way you determine particle hygroscopicity (I guess kappa is derived from CCNC data?).

**REPLY:**
Thank you for your question. The hygroscopicity parameter kappa was derived from VH-TDMA measurements (when instrument was running in H-TDMA mode). To specify this, we have included the following description to chapter 2.2.3:

The particle hygroscopicity parameter kappa (κ) was derived from VH-TDMA data following the κ-Köhler theory by Petters and Kreidenweis (2007):

$$\kappa = (GF^3 - 1) \cdot \left[ \frac{1}{S} \exp\left( \frac{4\sigma_s M_W}{RT\rho_W D_d GF} \right) - 1 \right] \qquad (1)$$

where $S$ is the saturation ratio; $\sigma_s$ - the surface tension of the solution; $M_W$ - the molecular weight of water; $R$ - the universal gas constant; $T$ - the temperature; $\rho_W$ - the density of water; and $D_d$ – particle dry diameter.

Since the new equation (Eq. 1) was introduced, we re-numbered the following equations accordingly.

3. Page 10, lines 283 to 287: Contamination control: Did you entirely rely on the CNC data or did you also include relative wind direction and velocity from the ships weather station?

**REPLY:**

Thank you for your question. A separate condensation particle counter (CPC) was used to measure total particle number concentration (PNC) with time resolution of 2 seconds (please note that in the text, chapter 3, time resolution was said to be 1 s, which is not correct). An exemplary data of PNC is presented in Fig. R1. It can be clearly seen that ship pollution heavily affects the momentary PNC. Even during NPF event, the maximum particle number concentration did not exceed 5000 particles $cm^{-3}$. Meanwhile, ship pollution resulted in order of magnitude higher concentrations. It is obvious that such increase must be related to ship exhaust pollution. Moreover, we have also compared PNC increase with a signal from Single Particle Soot Photometer ($SP^2$, operated inside the aerosol container by scientist from Alfred Wegener Institute. Data is not presented in the manuscript). After comparing the results from both $SP^2$ and CPC, we came to conclusion that CPC recorded sharp increase in PNC is a good indicator for ship-related pollution. Therefore, for online measurements only CPC and SMPS data was used for contamination inspection.

[Figure]

Fig. R1. Total particle number concentration during new particle formation event and contamination from ship.

That being said, automatic system (to measure relative wind direction) was installed together with high volume sampler to stop the pumps when wind came from a specific direction (related to ship exhaust). Similar approach was used by Huang et al. (2018), who performed her measurements inside same TROPOS measurement container onboard RV Polarstern. Berner impactor, on the other hand, did not have such system to prevent samples from contamination. However, in our work we did not use samples, which had high elemental carbon concentrations, indicating ship exhaust pollution.

Huang, S., Wu, Z., Poulain, L., van Pinxteren, M., Merkel, M., Assmann, D., Herrmann, H., and Wiedensohler, A.: Source apportionment of the organic aerosol over the Atlantic Ocean from 53° N to 53° S: significant contributions from marine emissions and long-range transport, Atmos. Chem. Phys., 18, 18043–18062, https://doi.org/10.5194/acp-18-18043-2018, 2018.

4. Page 10, lines 289 to 291: Data from Zeppelin Mountain Observatory: If these data are significant for your evaluation, I miss a more detailed discussion along with your results.

**REPLY:**

Thank you for your comment. Data from both Villum Research Station and Zeppelin Mountain Observatory was used only for supplementary information (to confirm whether NPF is a regional phenomenon) regarding NPF observed at RV Polarstern (as indicated in line 290: "data for visual inspection …"). We refrained ourselves from more detailed discussion, because we did not obtain/possess full data set regarding the NPF events. More detailed discussion would require a deeper look into data sets from Zeppelin Mountain Observatory and Villum Research Station, discuss the results with station responsible personnel/scientists etc., which is out of the scope of this study.

5. Page 18, lines 535 to 540: Estimating gaseous H2SO4 concentrations is crucial in order to describe nucleation and growth mechanism. If I understand aright, you used the tables and the given interpolation method presented by Yu (2010), where the basic input parameter is the observed nucleation rate (and not the growth rate)? Please clarify. In addition, you assume that because of the measured hygroscopicity parameter, only the binary system H2SO4-H2O is in the running and particularly not the tenary H2SO4-NH3-H2O system. In my view, it may still be worthwhile to envisage the latter option by comparison (see Napari et al., 2002).

**REPLY:**

Thank you for your comment/suggestion. For gaseous H2SO4 concentration, we reached out to Fangqun Yu (UAlbany), who was kind to introduce us to FORTRAN routine, which we used to estimate the assumed gas participating in NPF. The H2SO4 concentration was obtained by matching our calculated nucleation rate with corresponding modeled nucleation rate by varying H2SO4 concentration. When preparing the manuscript, we were also intended to use ternary nucleation mechanism to investigate our observed nucleation rates, however, Prof. Yu unfortunately did not have look up tables for ternary nucleation. Thus, we are grateful to referee #1, who pointed us to a valuable source of information, which we previously missed.

Based on our observed formation rates and Napari et al. (2002) parametrization, we estimated the concentrations of H2SO4 and NH3 (Fig. R2), which ranged from $1 \times 10^4$ to $5 \times 10^6$ cm$^{-3}$, and 0.1 to 100 ppt, respectively. Assuming Napari et al. (2002) used ppt as pptv for NH3, we see that such concentrations of ammonia can be indeed found in Artic region (Wentworth et al., 2016). In manuscript (Lines 560 – 570), we provided some sort of discussion regarding ammonia involvement in nucleation, which we expanded based on new results from the parametrization of ternary system, as suggested by referee #1. We have included the following text into manuscript:

Some studies (e.g. Croft et al., 2016; Köllner et al., 2017) identified that certain nitrogen-containing species such as ammonia and amines are linked to particle growth in the Arctic region. To test this, we investigate the formation rate of critical clusters using a parametrization of the ternary $H_2SO_4$-$NH_3$-$H_2O$ system, presented by Napari et al. (2002). That is, we adjust the concentrations of $H_2SO_4$ and $NH_3$ until we get the formation rate close to that of observed value. The estimated concentrations of $H_2SO_4$ and $NH_3$ varied from 1 x $10^4$ to 5 x $10^6$ $cm^{-3}$ and 0.1 to 100 ppt, respectively (see supplementary material). According to Wentworth et al. (2016), such concentrations of $NH_3$ can be indeed found in the Artic region. There is evidence that $H_2SO_4$-$NH_3$-$H_2O$ clusters are only partly neutralized under atmospheric conditions (e.g. Kurtén et al., 2007; Schobesberger et al., 2015). On the other hand, Asmi et al. (2010) reported that Aitken mode particles are somewhat more neutralized. Now, if we assume that the newly formed particles were partly neutralized by ammonia (as suggested by Giamarelou et al., 2016), we would expect particle hygroscopicity to be close to that of ammoniated sulfates. However, our observed HGF of 20 and 30 nm particles during both events were somewhat lower (e.g. 1.46 versus 1.64, Asmi et al., 2010).

References:

Asmi, E., Frey, A., Virkkula, A., Ehn, M., Manninen, H.E., Timonen, H., Tolonen-Kivimäki, O., Aurela, M., Hillamo, R. and Kulmala, M.: Hygroscopicity and chemical composition of Antarctic sub-micrometre aerosol particles and observations of new particle formation. Atmospheric Chemistry and physics, 10, 4253-4271, 2010.

Kurtén, T., Torpo, L., Sundberg, M.R., Kerminen, V.M., Vehkamäki, H. and Kulmala, M.: Estimating the NH3:H2SO4 ratio of nucleating clusters in atmospheric conditions using quantum chemical methods. Atmospheric Chemistry and Physics, 7, 2765-2773, 2007.

Schobesberger, S., Franchin, A., Bianchi, F., Rondo, L., Duplissy, J., Kürten, A., Ortega, I.K., Metzger, A., Schnitzhofer, R., Almeida, J. and Amorim, A.: On the composition of ammonia–sulfuric-acid ion clusters during aerosol particle formation. Atmospheric Chemistry and Physics, 15, 55-78, 2015.

Wentworth, G. R., Murphy, J. G., Croft, B., Martin, R. V., Pierce, J. R., Côté, J.-S., Courchesne, I., Tremblay, J.-É., Gagnon, J., Thomas, J. L., Sharma, S., Toom-Sauntry, D., Chivulescu, A., Levasseur, M., and Abbatt, J. P. D.: Ammonia in the summertime Arctic marine boundary layer: sources, sinks, and implications. Atmospheric Chemistry and Physics, 16, 1937–1953, 2016.

[Figure]

Fig. R2. Nucleation rate as a function of ammonia mixing ratio at T= 270 K and RH= 92% (according to parametrization by Napari et al. (2002). Total sulfuric acid concentration (in molecules per cm$^3$) is indicated for each curve. Dash-dot lines show the observed formation rate limits.

We have also included Fig. R2 to supplementary material.

6. Page 19, line 559: Sorry for nagging, but being a chemist, I have to note that ammonia is not an organic species (though it could be biogenic, if you mean this).

**REPLY:**
Thank you for your remark on this chemical incorrectness. Nagging is accepted. It is true that ammonia is not an organic species. However, ammonia is not exclusively biogenic either, even though biogenic sources are considered the biggest contributors for the Arctic atmosphere. Taking all into consideration, we changed the text in line 559 as follows:

Some studies (e.g. Croft et al., 2016; Köllner et al., 2017) identified that certain nitrogen-containing species such as ammonia and amines are linked to particle growth in the Arctic region.

7. Page 22, line 655 and eq. (7): The CCN increase due to NPF is defined as the difference between the CCN formed under background aerosol with diameter >100 nm and the CCN resulting from NPF. I do not understand why you used a cut-off of 100 nm defining background conditions. First, the model handles a bi-modal size distribution with geometric mean diameter and geometric standard deviation as input. Secondly, during all NPF events to be considered (referring to Fig. 2), particles did not grow above 50 nm. Thus, if at all, it would be reasonable defining a cut-off at around 50 nm instead of 100 nm for the background case. Please clarify!

**REPLY:**

Thank you for your comment. We agree that the definition of the CCN increase is somewhat not clear. In the following, we would like to clarify the line of thought when calculating the CCN increase due to NPF:

1. We calculate CCN number concentration resulting from parcel model using input values presented in Table 3. That is, we use those PNSDs, which is a result of new particle formation and subsequent growth. Model then outputs the CCN number concentration resulting from each mode (smaller particle mode at 13 to 44 nm; and larger particle mode at 101 to 194 nm). Total CCN number concentration is then a sum of CCN number from each mode. Please note that we calculate CCN number concentration resulting from 2 modes.

2. We then take total CCN number concentration (resulting from bi-modal size distribution, influenced by NPF) and divide it by the number of CCN resulting only from uni-modal distribution of >100 nm particles (from a different parcel model run, including only accumulation mode particles). We called these (>100 nm) particles as background aerosol, because from our observations it seems that it is always present in the Arctic atmosphere. In figure 2, one can see that average PNSD (PNSDs were taken from the cleanest episodes observed onboard RV Polarstern: 31 May 15:00 to 1 June 06:00; 14 June 15:00 to 15 June 15:00; 1 July 15:00 to 2 July 03:00; 5 July 21:00 to 6 July 12:00; 8 July 00:00 to 8 July 12:00) is indeed composed of two modes with geometric mean diameters at 22 and 132 nm.

   It might sound reasonable to calculate the increase in CCN number concentration with the reference to average background PNSD (which includes two modes). However, if looked at overall campaign PNSDs (not shown in the manuscript; available at request), in many cases the smaller particle mode is a result of new particle formation that happened either in the past days, or different location (and were transported to measurement site). Moreover, during NPF at 1 June, mode at <50 nm was not present at all. Another way would be to use PNSDs recorded just before each NPF event (Fig. R3). However, in this case, <50 nm particle mode is also frequently a result of the past NPFs. Therefore, it would be not entirely correct to include <50 nm mode if we want to estimate the increase in CCN number concentration due to NPF.

[Figure]

Fig. R3. Particle number size distributions (PNSD) measured prior new particle formation and campaign average PNSD (derived from the clean episodes). Please note that the mode at geometric mean diameter > 100 nm is present in all the cases, which cannot be said about ultrafine particle mode (PNSD before NPF at 1 June).

That being said, we have re-analyzed the increase in CCN number concentration in two different ways: 1) using the CCN number concentration resulting from campaign average PNSD (PNSD only from clean episodes); and b) using the CCN number concentration resulting from PNSD measured just before the NPF. Because we did not measure the hygroscopicity parameter kappa of 20 nm particles continuously, we used kappa value of 0.28 (average kappa value of 16-25 nm particles, measured during NPF). For accumulation mode particles, we assumed kappa of 0.33 (campaign average). In case 1, the number concentration of CCN (resulting from campaign average background PNSD) was 17 and 28 cm$^{-3}$, for updraft wind speeds of 0.1 and 3.2 m s$^{-1}$, respectively. This gave us the CCN number increase by 2 to 11 fold (versus 2 to 5 if old calculation from manuscript is used) for updraft wind speed of 0.1 m s$^{-1}$. For updraft wind speed of 3.2 m s$^{-1}$, the increase in CCN number was even higher – from 8 to 51 fold (versus 4 to 32 if old calculation from manuscript is used)

For case 2, the number concentration of CCN (resulting from PNSD just before NPF) depends on a specific PNSD, which was measured before every NPF event. Let us start with 1 June. The CCN number concentration, resulted from PNSD measured during 1 June (Fig. R3, black line) was 35 (for 0.1 m s$^{-1}$ updraft wind) and 40 (for 3.2 m s$^{-1}$ updraft wind) cm$^{-3}$. For lower updraft wind speed, parcel model did not show any increase in CCN number. However, for higher updraft wind, the CCN increase was 29 fold (versus 10 fold if old calculation method is used). For 18 June, the CCN increase was from 2 to 4 fold (same as in our old calculations) and from 6 to 20 fold (versus 6 to 32 fold in our old calculations) for updraft wind speeds of 0.1 and 3.2 m s$^{-1}$, respectively. And finally for 2 July, the CCN concentration increase was 5 fold in both updraft wind speed cases (same as in our old calculation). Please note that we did not include here calculations for 26 June NPF, because PNSD prior NPF was strongly affected by NPF on 24 and 25 June.

The main conclusions from such an exercise could be as follows:

1. Aerosol particle mode at geometric mean diameter of >100 nm was always present during the measurement campaign.
2. This cannot be said about <50 nm particle mode (see Fig. R3, PNSD before 1 June NPF).
3. It may seem sensible to use campaign averaged PNSD from clean episodes to define background PNSD, however, from our measurements, we noticed that in most cases the smaller particle mode is a result of NPF (either past days or transported from different location). Thus, it is not very precise to include nucleation mode particles in background aerosol definition when we try to estimate the increase in CCN number due to NPF.
4. Another approach could be using PNSD before NPF event to define background aerosol (same problem as in point 3).
5. We have calculated the increase in CCN concentration using two different PNSDs: for case one, campaign clean episode average PNSD; and PNSD recorded just before NPF event (point 3 and point 4 of these conclusions). We found that in case campaign average PNSD is assumed as background aerosol, the CCN increase is even higher (up to 11 fold (versus 5 fold; for updraft wind speed of 0.1 m s$^{-1}$). In case the PNSD prior NPF event was assumed as background aerosol, there was almost no difference compared to the results obtained using the methodology presented in the manuscript.
6. In both cases, we were able to show the CCN increase due to NPF.

Therefore, we tend to believe that our methodology to estimate the CCN number increase is correct (least wrong; we do not want to include <50 nm mode particles in the calculation, because most of the time these particles were the result of NPF during previous days). Nevertheless, we agree with referee #1 that some clarification is needed in the manuscript. For this, in the manuscript we have changed the following:

We define the CCN number concentration ($N_{CCN}$) increase due to particles created in the nucleation process as:

$$Increase\ of\ N_{CCN} = \frac{(N_{CCN,bp} + N_{CCN,NPF})}{N_{CCN,bp}}, \quad (8)$$

where ($N_{CCN,bp} + N_{CCN,NPF}$) is the number concentration of CCN resulting from the particles created in NPF event (calculated from bi-modal PNSD using parcel model; see Table 3 for simulation parameters) and $N_{CCN,bp}$ is the CCN number concentration resulting entirely from accumulation mode particles present during the NPF event (newly formed particle mode is suppressed in parcel simulation). For more detailed discussion about CCN increase calculation, please refer to supplementary materials.

Moreover, we included the above discussion to supplementary material.

8. Figure 2: This series of figures show a wealth of information that I could not really decipher in the present printed version. A magnification of the on-line version is necessary, but the resolution of the figures are somewhat poor. I would therefore recommend remedying this point.

**REPLY:**

Thank you for your recommendation. We realized that the quality of on-line version of figure 2 is truly poor. This may have happened due to the following: we have prepared the manuscript using MS Word. Meaning, Fig. 2 was imported and had to be resized to fit A4 format. Later, for submission to ACP, MS Word document was converted to pdf. We did not upload Fig. 2 as a separate vector graphics picture, resulting in poor resolution. We will make sure that in final version of manuscript, the highest quality figure will be included. For the moment, we attach Fig. 2 once again (in landscape orientation to maximize the resolution).

[Figure]

9. Figure 4: Fine, but what is about sea salt (Na+ and/or Cl-) data? These results are equally of some interest. For instance, a correlation with OC could give a hint whether part of the OC is primary aerosol that may be generated in conjunction with sea spray (just a mere suggestion for my part).

**REPLY:**

Thank you for the suggestion. We agree that looking at sea salt data is worthwhile, thus, we added a size-resolved sodium data to Figure 4 (see below). Sodium was mainly found on aerosol particles on Berner stages 3-5, while the majority of OM was determined on the submicron particles (Berner stages 1-3).

[Figure]

**Figure 4: Size-resolved atmospheric concentrations for ammonium, MSA, nss-sulfate, sodium, and OM for two sampling periods and the whole campaign average.**

Accordingly, we added this aspect to the text in the manuscript (Lines 435-447):

Sodium was mainly found on Berner stages 3 – 5. The sodium values for the sampling period from 25 to 28 June (Berner stage 4: 49 ng m-3) were quite similar to the average values, while the impactor samples from 29 May to 1 June showed much higher atmospheric concentrations (Berner stage 4: 386 ng m-3). This agrees well to previous studies, which show that atmospheric sea salt is mostly present in super-micron particles, while OM contributes strongly to the submicron particle composition (e.g. Müller et al., 2010). Previous works also suggest that OM is strongly enriched during the bubble bursting process (compared to sea salt) and therefore OM and sea salt are not transferred to the same extend from seawater to the aerosol particles (Keene et al., 2007; Quinn et al., 2015; Van Pinxteren et al., 2017). It is possible that increased sodium and OM, observed during NPF 1, is a result of sea spray, however, due to low sampling time resolution of Berner cascade impactor, we restrict ourselves from such conclusion. Moreover, please note that the increased values of sodium during this time period may be related to ship's proximity to open water (RV Polarstern reached the marginal ice zone only on 31 May), while the increase in OM could have happened later (e.g. 1 June), yet, included in the same sample. In chemical sample analysis, we did not find any positive correlation between OM/OC and sodium, concerning the different aerosol size classes. A more detailed chemical characterization of the aerosol particles during PS 106 cruise will be addressed in a separate publication.

Further changes:

We noticed that we missed to give a direct link between the Berner stages and their corresponding particle size. We fix this by including the following text into manuscript section 2.2.5:

The sampling of aerosol particles was conducted using five-stage low-pressure Berner impactors (Hauke, Austria) with a flow rate of 75 L $min^{-1}$, which was installed on the top of the observation deck facing the ocean at a height of ca. 25 m. Particles were collected in the size ranges 0.05 – 0.14 µm (stage 1), 0.14 – 0.42 µm (stage 2), 0.42 – 1.2 µm (stage 3), 1.2 – 3.5 µm (stage 4), and 3.5 – 10 µm (stage 5) aerodynamic particle diameter (50% cut-off) on aluminum foils as impaction substrates, which had been heated at 350 °C for at least 2 hours to reduce blank levels prior to sampling.

10. Page 16, line 453: It should be 5.4 nm h-1 (not 5.4 h-1).

**REPLY:**
Thank you. We fixed the noted typo.

11. Page 23, line 678: ...it can be concluded (not conclude).

**REPLY:**
Thank you. We fixed the noted typo.

**Response to comments of referee #2**

This manuscript describes an interesting and valuable set of measurements made aboard Polarstern in the region near Svalbard, during May to July 2017. The authors report measurements of neutral and charged clusters particle size (together covering a size range on <1 - 800nm), particle hygroscopicity and cloud condensation nuclei. Measurements of nucleation mode particle size and charged/neutral cluster abundance have been rarely made in Arctic regions, making this a unique and valuable data set that indirectly furthers our knowledge on the identity of species responsible for Arctic nucleation and initial growth of particles to Aitken mode sizes. The manuscript is well written and largely well organized. The following comments are intended to improve an already very good manuscript.

**REPLY:**

We thank to referee #2 for understanding the value of such studies. We also appreciate referee for his/her time, comments, and suggestions, which are intended to improve the manuscript. In the following, we would like to address the comments step-by-step.

**General Comments**

12. L65-73: This is an important section of the introduction, and at present is missing several of the relatively small amount of studies that exist. Since so few data sets exist, it is reasonable to cite them even if they are not all discussed in detail. Some that should be included are listed here: Freud et al, doi:10.5194/acp-17-8101-2017; Nguyen et al, doi: 10.5194/acp-16-11319-2016; Dall'Osto et al, doi: 10.5194/acp-19-7377-2019; Burkart et al, doi: 10.5194/acp-17-5515-2017; Burkart et al, doi: 10.1002/2017gl075671; Collins et al., doi: 10.5194/acp-17-13119-2017; Leaitch et al., doi: 10.12952/jour-nal.elementa.000017; Tremblay et al., doi: doi.org/10.5194/acp-19-5589-2019; These and others were recently reviewed in doi: 10.1029/2018rg000602.

**REPLY:**

Thank you for the comment. We have included the suggested references to manuscript. We have also re-wrote introduction to briefly overview the main findings of mentioned studies. Changes in introduction can be seen in yellow in a new manuscript version.

13. The discussion of ship deck observations in Section 2.1.1 may serve the reader better if they were incorporated into the larger discussion of NPF events in Section 3.1

**REPLY:**

Thank you for your suggestion. We merged Section 2.1.1 into Section 3.1. We agree that ship deck observations serve the reader better if they are incorporated into the discussion. Also, we noticed that some of the information is redundant. Changes are highlighted in yellow.

**3.1.1 NPF 1: 1 June**

[revised manuscript text omitted]

14. A detailed description of how pollution influence from the ship was reproducibly removed from the data set needs to be included in the Methods section. Was the influence quantified with a specific measurement (e.g., rapid variability in particle concentrations in a certain size range?)? Or, was a specific wind sector removed from the data? "Abrupt and short increases in particle number concentration recorded by a CPC" as cited in L286 as the method for filtering ship stack pollution; this should be elaborated in the methods section and detail is needed on how the authors determined that "most of the cases when the particle number increased tenfold" could be attributed to NPF. I strongly discourage the authors from including periods of ship pollution in Figure 2 (noted in L374-375). I disagree that "better representation of particle growth" (L1031-1032) warrants this.

**REPLY:**

Thank you for the comment. We would like to split the answer to referee #2 into several parts.

a. *Was the influence quantified with a specific measurement (e.g., rapid variability in particle concentrations in a certain size range?)?...“Abrupt and short increases in particle number concentration recorded by a CPC” as cited in L286 as the method for filtering ship stack pollution; this should be elaborated in the methods section and detail is needed on how the authors determined that “most of the cases when the particle number increased tenfold” could be attributed to NPF.*

**REPLY:**

Indeed, the pollution influence from the ship was quantified with a rapid variability in total particle number concentration. A separate condensation particle counter (CPC) was used to measure total particle number concentration (PNC) with time resolution of 2 seconds (please note that in the text, chapter 3, time resolution was said to be 1 s, which is not correct). An exemplary data of PNC is presented in Fig. R1. It can be clearly seen that ship pollution heavily affects the momentary PNC. Even during NPF event, the maximum particle number concentration did not exceed 5000 particles cm$^{-3}$. Meanwhile, ship pollution resulted in order of magnitude higher concentrations. It is obvious that such increase must be related to ship exhaust pollution. Moreover, we have also compared PNC increase with a signal from Single Particle Soot Photometer (SP$^2$, operated inside the aerosol container by scientist from Alfred Wegener Institute. Data is not presented in the manuscript). After comparing the results from both SP$^2$ and CPC, we came to conclusion that CPC recorded sharp increase in PNC is a good indicator for ship-related pollution. Therefore, for online measurements only CPC and SMPS data was used for contamination inspection.

Different methods exist to clean ambient background aerosol measurements from pollution plumes. For example, Kivekäs et al. (2014) used rolling window percentile to extract background concentrations from particle number size distributions effected by ship plumes. In our previous works, we used rolling window minimum method (see Kecorius et al., 2017). Nevertheless, in this work, we cleaned the online measurements by hand keeping 2 second time resolution CPC data as a reference. This way we are 100% certain that our analyzed results are ship contamination free. Software used to clean up the data was "Dafit" by Ries (2013).

b. *Or, was a specific wind sector removed from the data?*

**REPLY:**

Automatic system (to measure relative wind direction) was installed together with high volume sampler to stop the pumps when wind came from a specific direction (related to ship exhaust). Similar approach was used by Huang et al. (2018), who performed measurements inside same TROPOS measurement container onboard RV Polarstern. Berner impactor, on the other hand, did not have such system to prevent samples from contamination. However, we only used those samples, which had same order of OC with Digitel $PM_1$ samples. More detailed discussion regarding OC measurements using Digitel and Berner samplers will be provided in separate work. In the text, we addressed this limitation and added the following in chapter 3.4:

It must also be noted that there was no action taken (e.g. sampling interrupt dependent on specific wind sector) to reduce ship contamination for the size-segregated aerosol particle measurements. Thus, the contamination from the ship exhaust cannot be ruled out completely. However, the high concentrations of biogenic compounds like MSA and the presence of sodium on the aerosol particles suggested a strong marine influence to the particle composition.

[Figure]

Fig. R1. Total particle number concentration during new particle formation event and contamination from ship.

*c. I strongly discourage the authors from including periods of ship pollution in Figure 2 (noted in L374-375). I disagree that "better representation of particle growth" (L1031-1032) warrants this.*

**REPLY:**

Ship pollution was included only for visual representation. The data discussed in the manuscript is pollution free. Moreover, by removing data from Fig. 2 that is influenced by ship pollution, we would introduce blank gaps in contour plots. This would greatly diminish the visibility of particle new formation and subsequent growth, which is extended over time period of several days. The inclusion of pollution episodes in Fig. 2 does not obstruct the reader from seeing the processes happening during measurements.

To note the ship pollution influence onto our results, we have included a new section, 2.5 Contamination from ship exhaust, into the manuscript.

Huang, S., Wu, Z., Poulain, L., van Pinxteren, M., Merkel, M., Assmann, D., Herrmann, H., and Wiedensohler, A.: Source apportionment of the organic aerosol over the Atlantic Ocean from 53° N to 53° S: significant contributions from marine emissions and long-range transport, Atmos. Chem. Phys., 18, 18043–18062, https://doi.org/10.5194/acp-18-18043-2018, 2018.

Kecorius, S., Madueño, L., Vallar, E., Alas, H., Betito, G., Birmili, W., Cambaliza, M.O., Catipay, G., Gonzaga-Cayetano, M., Galvez, M.C. and Lorenzo, G., 2017. Aerosol particle mixing state, refractory particle number size distributions and emission factors in a polluted urban environment: Case study of Metro Manila, Philippines. Atmospheric environment, 170, pp.169-183.

Kivekäs, N., Massling, A., Grythe, H., Lange, R., Rusnak, V., Carreno, S., Skov, H., Swietlicki, E., Nguyen, Q.T., Glasius, M. and Kristensson, A., 2014. Contribution of ship traffic to aerosol particle concentrations downwind of a major shipping lane. Atmospheric Chemistry and Physics, 14(16), pp.8255-8267.

Ries, L.C., 2013. Dafit-a new work flow oriented approach for time efficient data preparation, validation and flagging of time series data from environmental monitoring. In EnviroInfo (pp. 651-656).

15. Further to comment (3) above, for the offline chemical analysis the sampling time was 72-144 hours (L223); how was sampling pollution from the ship minimized or avoided (e.g., by a wind switch)? The presence or absence of any such precautions should be stated, and if they were absent the possible implications should be discussed.

**REPLY:**

Thank you for the comment. As stated above, to note the ship pollution influence onto our results, we have included a new section, 2.5 Contamination from ship exhaust, into the manuscript.

16. A slightly more descriptive title might be helpful. For example, "during PS106 cruise" could be replaced by a few words describing the region of measurements. This would be helpful for readers not familiar with the region covered by the cruise.

**REPLY:**

Thank you for the suggestion. We agree with referee #2 that more descriptive title would benefit the reader. We have updated the title to "New particle formation and its effect on CCN abundance in the summer Arctic: a case study in Fram Strait and Barents Sea".

**Specific Comments**

1. L44: This statement might be best attributed to a paper such as Croft et al., Nature Communications, 2016, that makes a more direct connection to radiative forcing.

REPLY:
Thank you for the comment. We have included mentioned citation to manuscript.

2. L60-62: This statement is more attributable to Willis et al., Reviews of Geophysics, 2018 (https://agupubs.onlinelibrary.wiley.com/doi/full/10.1029/2018RG000602) and Abbatt et al., ACP, 2019 (https://www.atmos-chem-phys.net/19/2527/2019/acp-19-2527-2019.html).

REPLY:
Thank you for the comment. We have included mentioned citation to manuscript.

3. L63: Dall'Osto et al., Scientific Reports, 2018 corroborates these results for different regions and multiple stations (https://www.nature.com/articles/s41598-017-17343-9 and https://www.nature.com/articles/s41598-018-24426-8)

REPLY:
Thank you for the comment. We have included mentioned citation to manuscript.

4. L206: I assume that the CCN measurements were made on poly disperse aerosol, but this is not explicitly stated.

REPLY:
Thank you for the comment. Yes, CCN measurements were made only for poly disperse aerosol. Now we have included this statement in the manuscript.

5. L218-219: A large fraction of the organic aerosol may be semivolatile (e.g., Burkart et al, GRL, 2017). How might the semivolatile fraction be impacted by both heating of the inlet and long sampling times? Possibly biases introduced into these measurements should be discussed.

REPLY:
Thank you for the comment. Aerosol inlet for online measurements was heated to 30 °C to prevent ice formation and blockage of the inlet. Only very tip of the inlet was heated. Here, the residence time of aerosol in this heated area was 0.06 s. That is considerably shorter than aerosol spends inside instruments itself before being measured. For offline measurements, on the other hand, residence time is much higher – approx. 1.2 s (for Berner sampler). However, as stated in the text (Lines 219-220): The temperature difference between the ambient air at the impactor inlet and the sampled air after the conditioning unit did not exceed a value of 9 K. Taking into account the low outside temperatures, we do not expect a significant loss of semi volatiles due to this conditioning process (the maximum temperature that tube was heated to was 7 °C). The aerosol particles on the aluminum foils were not exposed to heating and sampled at the (cold) outside temperatures. We added this information in the manuscript (Line 193):

The losses due to evaporation of semi volatile compounds are expected to be minimal.

Furthermore, during sample transportation, samples were stored at -20 °C, thus, we do not expect losses during this stage.

6. L390-391: This is a very interesting observation and is in agreement with Tremblay et al., 2018 (doi above), could these data be included in the Supplement?

**REPLY:**

Thank you for the comment. We do not possess the observational data neither from Villum Research Station nor Zeppelin mountain Observatory. Nevertheless, for comparison purposes, particle number size distributions can be freely obtained from EBAS, developed and operated by the Norwegian Institute for Air Research (NILU, http://ebas.nilu.no/).

7. L425-429: Burkart et al., 2017 (doi above) came to similar conclusions.

**REPLY:**

Thank you for the comment. We have included mentioned citation.

8. L431-441 and Figure 4: Rather than referring to the stage number here I suggest the authors refer to the corresponding size range.

**REPLY:**

Thank you for the comment. We have updates the text and Fig. 4 as suggested by referee #2.

9. Related to comment (8), are any accumulation mode particle present during growth events? Do these larger modes also grow during NPF? Is sea salt present in accumulation mode sizes? Collins et al., 2017 (doi above) and Burkart et al., 2017 (doi above) observed growth of multiple modes at different rates showing that the species responsible for growing the larger mode was more semi volatile. If this is the case can the composition of the larger modes be connected to what is growing the smallest particles?

**REPLY:**

Thank you for the questions, which we would like to answer step-by-step:
1. *are any accumulation mode particle present during growth events?*

   From figure 2 (in manuscript) and figure R2 (see below), it can be seen that indeed there is an accumulation mode particles present during both particle formation and growth events.

2. *Do these larger modes also grow during NPF?*

   In three of four analyzed NPF events, we did not observe accumulation particle growth. Some accumulation particle growth could be seen during NPF event on 18 June. On 17 June, geometric mean diameter of accumulation mode particles was approx. 100 nm. In 32 hours (from 17 June 08:00 to 18 June 16:00), accumulation mode particle geometric mean diameter grew to 120 nm, that is 0.6 nm h$^{-1}$. Burkart et al. (2017) observed Aitken mode particle growth to sizes above 100 nm in parallel with nucleation particles. The growth rate was estimated to be 3.4 nm h$^{-1}$.

[Figure]

Fig. R2. Particle number size distribution during the new particle formation and growth events. Y-axis – mobility diameter (in nm), X-axis – time, color represents normalized particle concentration (dN/dlogD$_p$, cm$^{-3}$).

where S is the saturation ratio; $\sigma_s$ - the surface tension of the solution; $M_W$ - the molecular weight of water; R - the universal gas constant; T - the temperature; $\rho_W$ - the density of water; and $D_d$ – particle dry diameter.

**2.2.4 Cloud condensation particle counter (CCNC)**

The CCNC (model CCN-100 from Droplet Measurement Technologies, Roberts & Nenes, 2005) measured CCN number concentrations, subsequently at six different supersaturations (0.1, 0.15, 0.2, 0.3, 0.5 and 1%), where each supersaturation was sampled for 10 minutes. Hence an hourly average concentration at each supersaturation is available. The instrument was calibrated before and directly following the campaign using pure ammonium sulfate particles of known sizes, based on the ACTRIS protocol (Gysel & Stratmann, 2013). Only poly-disperse aerosol was sampled by CCNC.

**2.2.5 Offline chemical analysis**

The sampling of aerosol particles was conducted using five-stage low-pressure Berner impactors (Hauke, Austria) with a flow rate of 75 L min$^{-1}$, which was installed on the top of the observation deck facing the ocean at a height of ca. 25 m. Particles were collected in the size ranges 0.05 – 0.14 µm (stage 1), 0.14 – 0.42 µm (stage 2), 0.42 – 1.2 µm (stage 3), 1.2 – 3.5 µm (stage 4), and 3.5 – 10 µm (stage 5) aerodynamic particle diameter (50% cut-off) 
[revised manuscript text omitted]

**2.5 Contamination from ship exhaust**

During the cruise, ship exhaust occasionally disturbed measurements on board RV Polarstern This was mostly pronounced during the periods when ship was breaking the ice (rapid forward-backward direction change) and/or was drifting during sea experiments. Ship exhaust contamination can be seen in Fig. 2 contour plots as a sharp increase in particle number concentrations over the whole particle diameter range. The contamination from online measurements was removed manually. For this, we referred to total particle number concentration observed by separate CPC with 2 second time resolution. The comparison between total particle number concentration and the signal from single particle soot photometer (results are not shown here) confirmed that the total CPC indeed is able to observe sharp increase in particle number, which is related to ship exhaust (black carbon particles).

For off-line measurements, an automatic system (to measure relative wind direction) was installed together with high volume sampler to stop the pumps when the wind direction was associated with pollution sector. Similar approach was used by Huang et al. (2018). Berner impactor, on the other hand, did not have such system to

prevent samples from contamination. To avoid measurement artefacts, only samples with the same order of organic carbon as from high volume samplers were used for data analysis and discussion.

[revised manuscript text omitted]

It must also be noted that no action was taken (e.g. sampling interrupt dependent on specific wind sector) to reduce ship contamination for the size-segregated aerosol particle measurements. Thus, the contamination from the ship exhaust cannot be ruled out completely. However, the high concentrations of biogenic compounds like MSA and the presence of sodium on the aerosol particles suggested a strong marine influence to the particle composition.

The highest organic matter (OM) mass concentration were found on stage 2 (106 ng m$^{-3}$) and lowest - on stage 5 (39 ng m$^{-3}$). OM mass concentration for the period from 25 to 28 June strongly exceeded the average concentration, especially in the accumulation mode (218 ng m$^{-3}$ and 147 ng m$^{-3}$ for stages 2 and 3, respectively). For a time period from 29 May to 1 June the OM mass concentration ranged close to the average values.

Sodium was mainly found on Berner stages 3 – 5. The sodium values for the sampling period from 25 to 28 June (Berner stage 4: 49 ng m-3) were quite similar to the average values, while the impactor samples from 29 May to 1 June showed much higher atmospheric concentrations (Berner stage 4: 386 ng m-3). This agrees well to previous studies, which show that atmospheric sea salt is mostly present in super-micron particles, while OM contributes strongly to the submicron particle composition (e.g. Müller et al., 2010). Previous works also suggest that OM is strongly enriched during the bubble bursting process (compared to sea salt) and therefore OM and sea salt are not transferred to the same extend from seawater to the aerosol particles (Keene et al., 2007; Quinn et al., 2015; Van Pinxteren et al., 2017). It is possible that increased sodium and OM, observed during NPF 1, is a result of sea spray, however, due to low sampling time resolution of Berner cascade impactor, we restrict ourselves from such conclusion. Moreover, please note that the increased values of sodium during this time period may be related to ship's proximity to open water (RV Polarstern reached the marginal ice zone only on 31 May), while the increase in OM could have happened later (e.g. 1 June), yet, included in the same sample. In chemical sample analysis, we did not find any positive correlation between OM/OC and sodium, concerning the different aerosol size classes. A more detailed chemical characterization of the aerosol particles during PS 106 cruise will be addressed in a separate publication.

**4 Discussion**

**4.1 General overview**

Although NPF events in the high Arctic were reported by several studies, there are no observations, which use the same or equivalent measurement equipment as in this study, able to observe the dynamic changes of the smallest particles (formation and growth of >1.6 nm clusters). Because of this, we have also calculated the rate at which new particles appear at larger diameter (10 nm, $J_{10-}$). The values of so called apparent nucleation rates are more frequently reported in the literature. For example, in a several studies from the Svalbard region, GRs for 5 to 25 nm particles were reported to be from 0.1 to 0.6 nm h$^{-1}$, but in general ≤1.0 nm h$^{-1}$ (Ström et al., 2009; Giamarelou et al., 2016; Heintzenberg et al., 2017). The corresponding $J_{10-}$ values were in a range from 0.1 to 1.4 cm$^{-3}$ s$^{-1}$. Nieminen et al. (2018), on the other hand, reviewed NPF events based on long-term measurements and reported GRs for the Arctic region to be 1.1 – 1.2 nm h$^{-1}$ (for June – August time period). The reported formation rates were somewhat lower, 0.008 – 0.032 
[revised manuscript text omitted]
 certain nitrogen-containing species such as ammonia and amines are linked to particle growth in the Arctic region. To test this, we investigate the formation rate of critical clusters using a parametrization of the ternary $H_2SO_4$-$NH_3$-$H_2O$ system, presented by Napari et al. (2002). That is, we adjust the concentrations of $H_2SO_4$ and $NH_3$ until we get the formation rate close to that of observed value. The estimated concentrations of $H_2SO_4$ and $NH_3$ varied from 1 x $10^4$ to 5 x $10^6$ cm$^{-3}$ and 0.1 to 100 ppt, respectively (see supplementary material). According to Wentworth et al. (2016), such concentrations of $NH_3$ can be indeed found in the Artic region. There is evidence that $H_2SO_4$-$NH_3$-$H_2O$ clusters are only partly neutralized under

atmospheric conditions (e.g. Kurtén et al., 2007; Schobesberger et al., 2015). On the other hand, Asmi et al. (2010) reported that Aitken mode particles are somewhat more neutralized. Now, if we assume that the newly formed particles were partly neutralized by ammonia (as suggested by Giamarelou et al., 2016), we would expect particle hygroscopicity to be close to that of ammoniated sulfates. However, our observed HGF of 20 and 30 nm particles during both events were somewhat lower (e.g. 1.46 versus 1.64, Asmi et al., 2010). Similar hygroscopicity of ultrafine particles (HGF = 1.38 for 15 nm particles) in the Arctic was observed by Zhou et al. (2001). However, authors excluded the water-sulfuric acid nucleation as a source of such particles because <50 nm particles did not appear to be composed neither of sulfuric acid nor ammonium sulfate. Kim et al. (2016) measured the hygroscopicity of nanoparticles produced from homogeneous nucleation in the CLOUD experiment. If compared to CLOUD experiment results, the measured hygroscopicity of 20 nm particles during Event 1 was closest to the results of experiment, during which sulfuric acid and dimethylamine (DMA) concentrations were $7.6 \times 10^6$ molecules cm$^{-3}$ and 23.8 ppt, respectively. With that being said, experiment with sulfuric acid ($15.1 \times 10^6$ molecules cm$^{-3}$) and organics produced from α-pinene ozonolysis (420 ppt) resulted in 15 nm particles with HGF = 1.33, which is identical to those observed during Event 4.

To conclude, one can only assume that during Events 1 and 4, the NPF was initiated by sulfuric acid. The involvement of ammonia in new particle formation although is possible, cannot be proved by this work. The organics of marine-origin could have been involved in particle growth to some extent. However, low (compared to campaign average) organic matter concentrations, observed by offline chemical analysis, oppose to aforesaid conclusion. The hypothesis that NPF is driven by sulfuric acid can be supported by the results of neutral cluster and ion number size distribution and hygroscopicity measurements of nucleation mode particles.

**4.2.2 NPF 2 and 3**

[revised manuscript text omitted]